



# Effects of Near-Source Coagulation of Biomass Burning Aerosols on Global Predictions of Aerosol Size Distributions and Implications for Aerosol Radiative Effects

Emily Ramnarine[1], John K. Kodros[1], Anna L. Hodshire[1], Chantelle R. Lonsdale[2], Matthew J. Alvarado[2], Jeffrey R. Pierce[1]

[1]Colorado State University, Department of Atmospheric Science, Fort Collins, CO, 80523, USA
[2]Atmospheric and Environmental Research, Lexington, MA, 02421, USA

*Correspondence to*: Emily Ramnarine (emily.ramnarine@colostate.edu)

**Abstract.** Biomass burning is a significant global source of aerosol number and mass. In fresh biomass burning plumes, aerosol coagulation reduces aerosol number and increases the median size of aerosol size distributions, impacting aerosol radiative effects. Near-source biomass burning aerosol coagulation occurs at spatial scales much smaller than the grid boxes of global and many regional models. To date, these models ignore sub-grid coagulation and instantly mix fresh biomass burning emissions into coarse grid boxes. A previous study found that the rate of particle growth by coagulation within an individual smoke plume can be approximated using the aerosol mass emissions rate, initial size distribution median diameter and modal width, plume mixing depth, and wind speed. In this paper, we use this parameterization of sub-grid coagulation in the GEOS-Chem-TOMAS global aerosol microphysics model to quantify the impacts on global aerosol size distributions, the direct radiative effect, and the cloud-albedo aerosol indirect effect.

We find that inclusion of biomass burning sub-grid coagulation reduces the biomass burning impact on the number concentration of particles larger than 80 nm (a proxy for CCN-sized particles) by 37% globally. This CCN reduction causes our estimated global biomass burning cloud-albedo aerosol indirect effect to decrease from -76 to -43 mW m$^{-2}$. Further, as sub-grid coagulation moves mass to sizes with more efficient scattering, including it increases our estimated biomass burning all-sky direct effect from -224 to -231 mW m$^{-2}$ with assumed external mixing and from -188 to -197 mW m$^{-2}$ with assumed internal mixing with core-shell morphology. However, due to differences in fire and meteorological conditions across regions, the impact of sub-grid coagulation is not globally uniform. We also test the sensitivity of the impact of sub-grid coagulation to two different biomass burning emission inventories, to various assumptions about the fresh biomass burning aerosol size distribution, and to two different timescales of sub-grid coagulation. The impacts of sub-grid coagulation are qualitatively the same regardless of these assumptions.



## 1 Introduction

Atmospheric aerosol particles, including those from biomass burning, impact the climate system directly by scattering and absorbing radiation and indirectly by influencing cloud properties (Bauer et al., 2010; Bond and Bergstrom, 2006; Bond et al., 2013; Hodshire et al., 2018; Kodros et al., 2015; Kodros et al., 2016; Kodros and Pierce, 2017; Pierce et al., 2007; Reid

et al., 2005; Twomey, 1974; Weigum et al., 2016). In this paper, biomass burning includes wildfires, prescribed burns, and agricultural burning, but not residential or industrial biofuel use. Emissions from biomass burning include organic aerosol (OA), black carbon (BC), and inorganic particles, as well as aerosol precursor vapors such as sulfur dioxide and volatile organic compounds (e.g. Akagi et al., 2011). The largest biomass burning emissions occur over tropical Africa, South America, and Southeast Asia, but substantial emissions also occur in temperate and boreal forests (Bond et al., 2013; van der

Werf et al., 2017; Wiedinmyer et al., 2011). Biomass burning smoke concentrations are spatially and temporally heterogeneous throughout most regions (Rodhe et al., 1972; Bond et al., 2013), and biomass burning aerosol may be transported thousands of kilometers downwind, potentially affecting areas far from the emitting fires (e.g., Val Martin et al., 2006). Bond et al. (2013) estimated that biomass burning makes up 66% of primary OA mass emissions and 37% of BC mass emissions, globally.

Aerosol emissions from biomass burning impact climate in a variety of ways. In the direct radiative effect (DRE), scattering and absorption of shortwave radiation by biomass burning particles leads to an increase or decrease in planetary albedo, respectively, resulting in a negative (cooling tendency) or positive (warming tendency) radiative effect, respectively. OA from biomass burning plumes predominantly scatters solar radiation, while BC predominantly absorbs (Bond et al., 2013).

The efficiency of this scattering and absorption depends on the size and mixing state of the particle (Bond and Bergstrom, 2006; Kodros et al, 2015; Seinfeld and Pandis, 2016; Alvarado et al., 2016). In the cloud-albedo aerosol indirect effect (AIE), aerosols acting as cloud condensation nuclei (CCN) lead to an increase in cloud droplet number concentration (CDNC) and the shortwave albedo of clouds (Twomey, 1974). The ability of an aerosol to act as a CCN depends on its concentration, size, and solubility (Petters and Kreidenweis, 2007). As biomass burning plumes age, the aerosol size

distribution evolves due to coagulation, condensation, and evaporation (Bian et al., 2017; Hodshire et al., 2018; Sakamoto et al., 2016); and this evolution impacts the aerosol radiative effects (Bauer et al., 2010; Pierce et al. 2007; Reid et al., 2005).

In this paper, we focus on coagulation in biomass burning plumes. Coagulation is the aggregation of particles upon collision, combining two particles into one larger particle. The rate of coagulation depends on particle size and concentration, and is

fastest when there is a high concentration of particles and a large spread in the sizes of those particles (Seinfeld and Pandis, 2016). The coagulation rate is proportional to the square of the particle number concentration for fixed particle sizes, and hence is strongly dependent on the number concentration (Seinfeld and Pandis, 2016). As coagulation occurs, there is a reduction in the number concentration of smaller particles, leading to an overall reduction of particle number and narrowing



of the size distribution (Seinfeld and Pandis, 2016). As biomass burning plumes generally have spatial scales much smaller than the width of global and regional model grid boxes, these models cannot explicitly resolve smoke plumes. Biomass burning particles are thus instantly mixed throughout the gridbox volume upon emission. This instantaneous mixing in the grid boxes dilutes the concentration of particles and likely causes an underprediction of coagulation rates, leading to an

overprediction of biomass burning number concentrations and errors in the size distribution of these particles (Stuart et al., 2013).

The impact of coagulation on the size distribution of aerosols in biomass burning smoke plumes was explored by Sakamoto et al. (2016), where they developed a physically intuitive coagulation parameterization for individual smoke plumes. To

develop this parameterization, Sakamoto et al. (2016) simulated individual smoke plumes using a large-eddy simulation model with size-sectional aerosol coagulation and no other aerosol processes. The simulated data from these model runs was used to fit equations for changing median diameter and modal width with plume aging. These equations show that the rate of coagulation of a single fire plume can be approximated using the mass emissions rate of biomass burning aerosol (the product of emissions flux and fire area), initial size distribution (median diameter and modal width), plume mixing depth,

and wind speed. Intuitively, more-concentrated emissions, larger area fires, smaller wind speed, or smaller mixing depth lead to an increased rate of coagulation. This increased rate of coagulation is represented by a larger median diameter and smaller modal width for equivalently aged smoke. Sakamoto et al. (2016) further showed that the parameterization is more skillful at predicting measured aged median diameter and modal width values than assuming constant values.

In this paper, we use a global aerosol microphysics model to explore how sub-grid coagulation of biomass burning emissions impacts global aerosol size distributions, the DRE, and the cloud-albedo AIE. We also quantify the sensitivity of biomass burning radiative effects to changing the mixing-state assumption, initial aerosol size distribution, biomass burning emissions inventory, and the timescale of the sub-grid coagulation. Section 2 describes our methods. Section 3 presents the results of our model simulations and includes a discussion of changes to the size distribution globally, details on changes to

the size distribution in two representative locations, an analysis of the changes to the radiative effects under the conditions of our sensitivity studies, and a consideration of limitations of this study. Our conclusions are summarized in Sect. 4.

## 2 Methods

### 2.1 Model Overview

We use the global chemical-transport model GEOS-Chem version 10.01 (http://acmg.seas.harvard.edu/geos/) with 4°×5°

horizontal resolution and 47 vertical levels. Our simulations use Goddard Earth Observing System model version 5 (GEOS5) meteorological re-analysis fields. Because the meteorology is offline, changes to aerosol concentrations do not feedback to affect meteorology, and so all cases here have identical meteorology. Our simulations use meteorology for the year 2010



with one month of spin-up not used in analysis. GEOS-Chem is coupled with the TwO-Moment Aerosol Sectional (TOMAS) microphysics model (Adams and Seinfeld, 2002; Kodros and Pierce, 2017; Trivitayanurak et al., 2008). This version of TOMAS has 15 size sections corresponding to dry particle diameters ranging from approximately 3 nm to 10 μm, and includes tracers for sulfate, sea salt, OA, BC, dust, ammonia and particle-phase water. OA mass is assumed to be 1.8

times that of organic carbon as a central value from Philip et al. (2014). TOMAS explicitly simulates both aerosol number and mass within each size section. Detailed descriptions of microphysical processes in TOMAS are described in Adams and Seinfeld (2002), Lee and Adams (2012), and Lee et al. (2013).

We test model simulations with biomass burning emissions from the Global Fire Emissions Database version 4 (GFED; van

der Werf et al., 2017) and the Fire INventory from NCAR (FINN; Wiedinmyer et al., 2011). GFED has a resolution of 0.25°×0.25° spatial resolution and daily temporal resolution. It uses an adapted terrestrial carbon cycle model (Carnegie-Ames-Stanford Approach; CASA model) to estimate fuel combustion per unit area. CASA uses MODIS vegetation and land cover products, ERA-interim meteorology, and ERA-interim soil moisture as inputs. The CASA-estimated fuel combustion per unit area is combined with the MODIS burned area product and emission factors from Akagi et al. (2011) to calculate

emissions (van der Werf et al., 2017). FINN uses the MODIS thermal anomaly product to detect daily fire emissions with a resolution of one square kilometer, and uses the MODIS vegetation product for land cover to determine fuel loading and fraction of biomass burned. In FINN version 1.5 (FINNv1.5), these estimates of mass of biomass burned are converted into mass emissions of a variety of species for each fire using emission factors (Wiedinmyer et al., 2011). The use of FINNv1.5 in this study is discussed further in Sect. 2.2. In FINN version 1 used in GEOS-Chem (FINNv1), the emissions of carbon

dioxide from each individual fire has been gridded to 0.25°×0.25° spatial resolution and other emissions are determined using emission ratios (relative to carbon dioxide) based on vegetation type (Wiedinmyer et al., 2011).

Because GFED and FINN are derived differently, their subsequent emission fields are also different. FINN, being based on active fires and intended for near-real-time use, may be better at capturing variability in regions with many small fires, but

does not take into account variability in fuel consumption or fire area at the sub-biome scale (Reddington et al, 2016; Wiedinmyer et al., 2011). Both GFED and FINN are derived from satellite products, which may lead to missing emissions from very small fires (Huang et al., 2018). In studies comparing multiple fire emission inventories that include GFED and FINN, FINN tends to be an outlier in that it does not have a statistically significant cross-correlation (spatial and temporal) with most other inventories, while all other inventories are significantly correlated with each other (Shi and Matsunaga,

2017; Shi et al., 2015). Hence, we choose to use GFED fire emissions as the default in this paper. For completeness, we include figures using the FINN emissions in the supplement.



**Table 1: Simulation names and descriptions of GEOS-Chem parameters which change depending on the simulation. In the naming, '*SubCoag*' refers to the inclusion of sub-grid coagulation and '*noSubCoag*' indicates the exclusion of sub-grid coagulation. The default size distribution has an emitted median diameter of 100 nm and an emitted modal width of 2. '*D150*' in the name indicates that the median diameter is increased to 150 nm. '*s1.6*' indicates that the modal width is decreased to 1.6 (with the '*s*' coming from 'sigma'). The default biomass burning emissions inventory is GFED, and simulations using FINNv1 instead include '*FINN*' in their names. The default amount of time spent undergoing sub-grid coagulation is 24 hours and the simulation with only 12 hours spent aging has a '*_12h*' suffix.**

| Simulation | Biomass Burning Emissions Inventory | Emitted number median diameter ($D_{pm0}$ ; nm) | Emitted number modal width ($\sigma_0$) | Time spent undergoing sub-grid coagulation (*time*; hours) |
|---|---|---|---|---|
| *noBB* | none | -- | -- | -- |
| *noSubCoag* | GFED | 100 | 2 | 0 |
| *SubCoag* | GFED | 100 | 2 | 24 |
| *SubCoag_12h* | GFED | 100 | 2 | 12 |
| *D150_noSubCoag* | GFED | 150 | 2 | 0 |
| *D150_SubCoag* | GFED | 150 | 2 | 24 |
| *s1.6_noSubCoag* | GFED | 100 | 1.6 | 0 |
| *s1.6_SubCoag* | GFED | 100 | 1.6 | 24 |
| *noSubCoag_FINN* | FINNv1 | 100 | 2 | 0 |
| *SubCoag_FINN* | FINNv1 | 100 | 2 | 24 |
| *D150_noSubCoag_FINN* | FINNv1 | 150 | 2 | 0 |
| *D150_SubCoag_FINN* | FINNv1 | 150 | 2 | 24 |
| *s1.6_noSubCoag_FINN* | FINNv1 | 100 | 1.6 | 0 |
| *s1.6_SubCoag_FINN* | FINNv1 | 100 | 1.6 | 24 |



## 2.2 Biomass-burning emissions size and sub-grid coagulation in GEOS-Chem-TOMAS

Table 1 provides and overview of the simulations performed in this study and that will be referenced in this section. Fresh biomass burning aerosol emission sizes can generally be simplified to a lognormal mode, but the parameters of this mode vary with fire characteristics (Janhäll et al., 2010). This emitted lognormal mode is not specified in either of the emission inventories used. In GEOS-Chem-TOMAS, the default biomass burning emissions have an emitted number median diameter of 100 nm and modal width of 2. Janhäll et al. (2010) provides a review of measurements of fresh and aged smoke, and these default values already in use in GEOS-Chem-TOMAS are on the small end of the median diameters, which range from 100 nm to 141 nm, and the large end of modal widths, which range from 1.5 to 1.91, of the fresh plumes studied. To address uncertainty in emitted aerosol size distributions, we include sensitivity simulations varying the initial emission median diameter and modal width of the biomass burning aerosol (see Table 1). One set of simulations increases the emitted median diameter from 100 nm (*noSubCoag*) to 150 nm (*D150_noSubCoag*) with a constant a modal width of 2. The second set of simulations decreases the emitted modal width from 2 to 1.6 (*s1.6_noSubCoag,* where the 's' represents 'sigma' for modal width) with a constant median diameter of 100 nm. Each of these simulations has a counterpart with sub-grid coagulation included (*SubCoag*, *D150_SubCoag*, and *s1.6_SubCoag*, respectively). These simulations are all run with GFED emissions (as that is the default), but there are corresponding simulations with FINN emissions (*noSubCoag_FINN, D150_noSubCoag_FINN, s1.6_noSubCoag_FINN, SubCoag_FINN, D150_SubCoag_FINN,* and *s1.6_SubCoag_FINN).* We compare these simulations to a model simulation with no biomass burning emissions of particles or gases (*noBB*). By choosing this variety of simulations, we represent a range of fresh biomass burning plume emission size distributions.

To represent the evolution of the aerosol size distribution due to in-plume sub-grid coagulation in the *SubCoag* simulations, we use the parameterization developed in Sakamoto et al. (2016). The parameterization is for individual, non-overlapping plumes:

$$D_{pm} = D_{pm0} + 84.56 \left[ \frac{Emissions\ Rate}{(Wind\ Speed)(Mixing\ Depth)} \right]^{0.4191} (Time)^{0.4870} , \qquad (1)$$

$$\sigma = \sigma_0 + 0.2390 \left[ \frac{Emissions\ Rate}{(Wind\ Speed)(Mixing\ Depth)} \right]^{0.1889} (Time)^{0.3540} (1.2 - \sigma_0) , \qquad (2)$$

where $D_{pm}$ is the median diameter after in-plume coagulation (nm), $D_{pm0}$ is the initial number median diameter before in-plume coagulation (nm), $\sigma$ is the modal width after in-plume coagulation, and $\sigma_0$ is the initial modal width before in-plume coagulation. $Time$ is the amount of time spent undergoing in-plume coagulation (min). $Emissions\ Rate$ is the mass emissions rate of primary aerosol from a fire (kg min[-1]), which is OA and BC emissions in our simulations, as it was in Sakamoto et al. (2016). $Wind\ Speed$ and $Mixing\ Depth$ are the meteorological wind speed (m min[-1]) and the depth that the smoke plume mixes through (m), respectively. Figure S1 shows that when the sub-grid coagulation parameterization (Eqs. 1




and 2) is used offline the downwind median diameter is larger than the emitted median diameter and the downwind modal width is smaller than the emitted modal width. This change is larger for fires with higher emission rates, but there is some variability because local meteorology also plays a role.

The parameterization is included in GEOS-Chem with some limitations. As GEOS-Chem currently emits all biomass burning emissions into the boundary layer, mixing depth is approximated by the planetary boundary layer depth, a meteorology field. As better data on injection heights and mixing depth are developed and/or included into GEOS-Chem they may be coupled into our use of the parameterization. We acknowledge that keeping all emissions in the boundary layer is a limitation of this study (Paugam et al., 2016); however, Rémy et al. (2003) found that most injection heights are within

the planetary boundary layer. In our simulations, the minimum mixing depth was 10 m, the minimum wind speed was 2 m min$^{-1}$, and the minimum emissions rate was 1 kg min$^{-1}$. These minimums are defined to avoid getting unrealistic values out of the parameterization. We use the 10 m wind speed. We chose to age the plume for 24 hours as that is approximately the time it takes for air to cross a 4°×5° gridbox in the boundary layer, assuming that the plume started at one edge and dilutes all the way across. In reality, the fire could start anywhere within the grid box, and we acknowledge that this choice of

timescale is somewhat arbitrary, but the roughly square-root dependence of the size distribution on time means the parameterization is only weakly sensitive to the choice of timescale. To test the sensitivity to this 24-hour assumption we include an additional simulation (*SubCoag_12h*) where conditions are similar to *SubCoag* but with 12 hours instead of 24 hours of aging.

To implement the sub-grid coagulation parameterization into GEOS-Chem-TOMAS, we take the number of fires per gridbox from FINNv1.5 into GEOS-Chem via the Harvard-NASA Emissions Component (HEMCO), regardless of whether FINNv1 or GFED is used for fire emissions. When using the Sakamoto parameterization in GEOS-Chem-TOMAS, all fires within a single gridbox are treated as the same in that the emissions are distributed evenly, leading to the same $D_{pm}$ and $\sigma$ for all fires in that gridbox. In doing so, we assume that there is no overlap of the fire plumes (a shortcoming that we will discuss later).

This single lognormal mode is applied to the gridbox to determine how BC and OC number and mass are distributed across sizes. In gridboxes where GFED or FINNv1 has fire emissions and FINNv1.5 does not have any fires, one fire is assumed. Figure S2 shows through offline calculations that using the Sakamoto parameterization in this way results in approximately the same result as using the Sakamoto parametrization for each individual fire and then averaging the resulting $D_{pm}$ and $\sigma$ over the 4°×5° grid, justifying our method of applying this parameterization to a gridded model. Over the Sahara, the ice

sheets, and the ocean there are no fires and therefore no effect of sub-grid coagulation.



## 2.3 Modeling radiative impacts of changes made to biomass burning emissions

To estimate the radiative impacts of these simulations, we use an offline version of the Rapid Radiative Transfer Model for Global Climate Models (RRTMG). Implementation of RRTMG with GEOS-Chem-TOMAS simulations is described in detail in Kodros et al. (2016). When estimating the DRE, we assume either a fully internal or an external mixing state. For the internal mixture, we assume a core-shell morphology where black carbon forms a spherical core and other species form a homogeneously mixed shell around that core, remaining spherical (Jacobson, 2001). An external mixture assumes that organic carbon and black carbon remain separate, each forming their own set of spherical particles. These mixing-state assumptions are idealizations used here to provide bounds on the magnitude of the DRE where core-shell mixing (in which the shell acts as a lens to enhance the warming in the core) is the warmest forcing and external mixing is the coolest forcing. When estimating the AIE, we assume all aerosol species are mixed internally within each size bin to calculate $\kappa$ (the hygroscopicity parameter; Petters and Kreidenweis (2007)) with the exception of fresh black carbon, which becomes internally mixed on a fixed e-folding timescale of 1.5 days (Pierce et al., 2007). The fresh black carbon is assumed to be externally mixed with a $\kappa$ of zero and hence does not activate (Pierce et al., 2007). Details of the calculation of CDNC, cloud properties, and AIE are discussed in Kodros et al. (2016). The offline calculation of aerosol radiative effects described here uses monthly mean aerosol concentrations and meteorological inputs, which we note as a limitation of this study.

## 3 Results

### 3.1 Impact of biomass burning on aerosol mass

Figure 1 shows the spatial distribution of the biomass burning impacts on annually averaged simulated OA and BC mass concentrations (the difference between the *noSubCoag* and *noBB* simulations, both defined in Table 1). Biomass burning makes up most of the column OA and BC mass in our simulations over areas with significant biomass burning emissions and/or few other sources. These regions include the Amazon in South America, the Congo in Africa, and some regions of the boreal forests in North America and Siberia. Biomass burning aerosol also accounts for most of the column OA and BC mass in remote areas downwind of biomass burning emissions, including most of the southern hemisphere. Figure S3 provides the same analysis as Fig. 1 but for the FINNv1 emissions inventory. It shows that FINNv1 also has biomass burning accounting for a significant amount of OA and BC mass over major biomass burning areas and downwind. FINN has 52 Tg of OA+BC emissions over the course of the 2010 year, while GFED has 60 Tg of emissions. Because FINNv1 has a lower mass of emissions, the increase in OA and BC mass over major biomass burning regions and downwind are not as high as they are with GFED emissions.



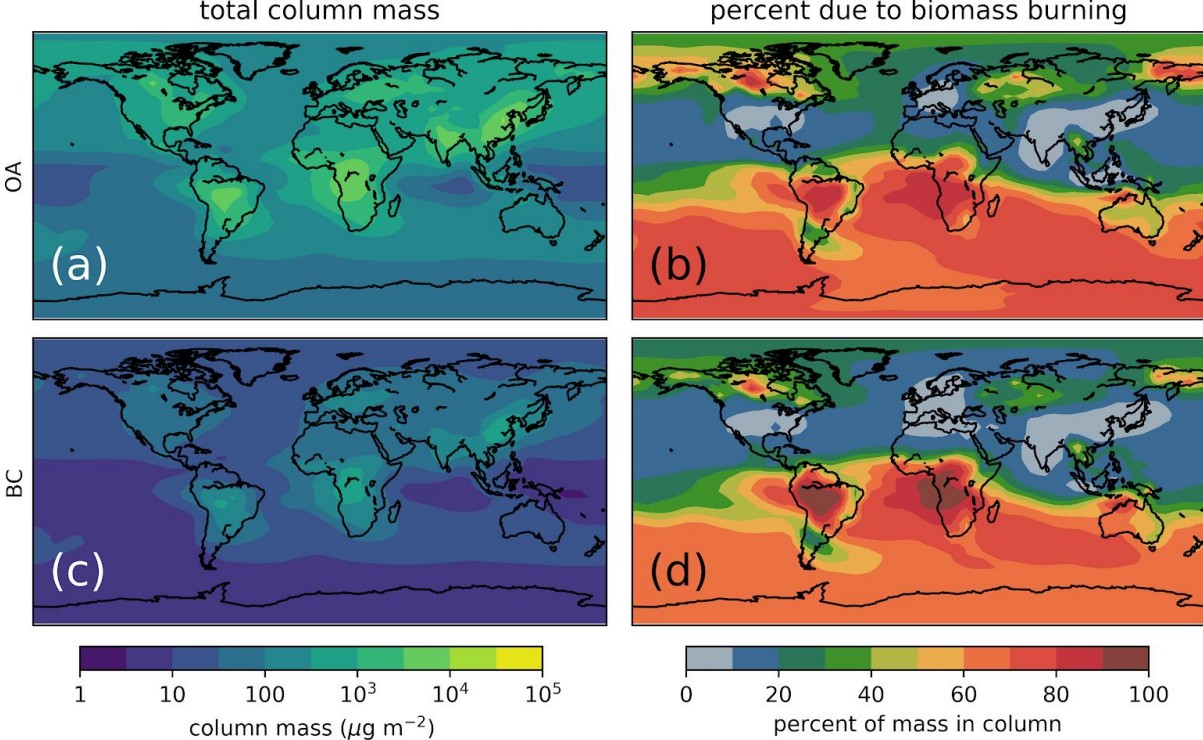

**Figure 1: Effect of biomass burning on annually averaged total column OA and BC mass concentrations. The left side shows the total column mass concentration of (a) OA and (c) BC in the simulations with GFED biomass burning emissions (i.e., *noSubCoag*). The right side shows the  percent of the mass in the column that is due to biomass burning emissions for (b) OA and (d) BC by taking the difference between the *noSubCoag* and *noBB* simulations.**

### 3.2 Ambient size distribution sensitivity to biomass burning

Figure 2 shows the change in the spatial pattern of aerosol number concentration due to biomass burning for both the standard coagulation scheme without sub-grid coagulation (*noSubCoag - noBB*) and the scheme with the sub-grid coagulation parameterization included (*SubCoag - noBB*). Shown are the changes in number concentrations for particles with diameters larger than 10 nm (N10) and particles with diameters larger than 80 nm (N80, a proxy for CCN-sized particles). The greatest increases in N10 and N80 (over a doubling in some areas) for both coagulation schemes occur over the regions with the largest biomass burning emissions: the Amazon, the Congo, Southeast Asia, and the boreal forests in North America and eastern Siberia. There are also increases downwind of these high-emission areas. Many regions, such as central Asia and the remote oceans, show decreases in particle number due to the inclusion of biomass burning. The increase in primary particle number in biomass burning source regions increases the condensation sink, which leads to a reduction in new-particle formation and growth and an increased coagulational loss of new particles, leading to lower number concentrations





away from biomass burning regions. Similar remote-region number decreases were also seen when adding primary-particle sources in Kodros et al. (2015; 2016). Figure S4 shows that FINNv1 emissions have a smaller percent increase in N10 and N80 over biomass burning areas than GFED emissions because FINNv1 emits less particle mass, as discussed in Section 3.1. Because of the lower emissions in FINN, there is also a smaller decrease downwind.

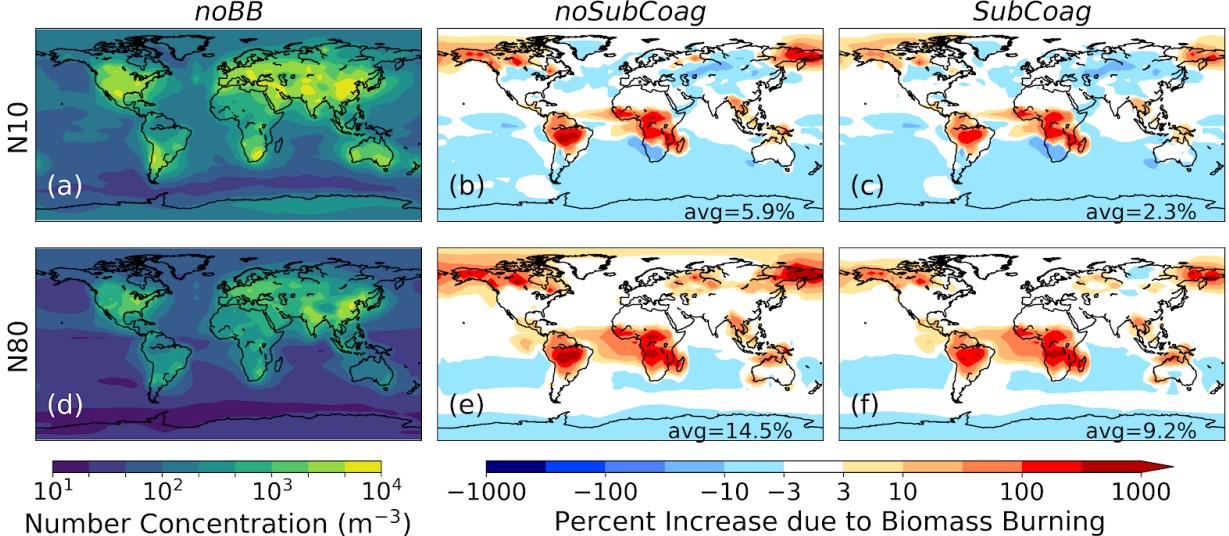

**Figure 2: Effect of biomass burning on surface-level number concentration of particles larger than 10 nm (a, b, c) and 80 nm (d, e, f). Panels (a) and (d) show the absolute number concentration for the *noBB* simulation. Panels (b) and (e) show the percent increase due to GFED biomass burning emissions from the *noBB* simulation to the *noSubCoag* simulation. Panels (c) and (f) show**

**the percent increase due to GFED biomass burning emissions from the *noBB* simulation to the *SubCoag* simulation. The number in the bottom right of each panel is the global mean percent increase due to biomass burning.**

When including sub-grid coagulation (Fig. 2, panels (c) and (f)), there are lower N10 concentrations over emissions areas due to the sub-grid coagulation as compared to the standard coagulation scheme without sub-grid coagulation (Fig. 2, panels

(b) and (e)), but the mass remains approximately the same (emissions mass is the same but scavenging may be different due to changes in particle sizes). Because of this, the global, annual-average percent increase in N10 due to biomass burning is reduced from 5.9% without sub-grid coagulation to 2.3% with sub-grid coagulation. Likewise, the global-, annual-average percent increase in N80 due to biomass burning is reduced from 14.5% without sub-grid coagulation to 9.2% with sub-grid coagulation. Because coagulation removes smaller particles more efficiently, the decrease in N10 is more dramatic than the

decrease in N80. The same effect of sub-grid coagulation can be seen in the simulations with FINNv1 emissions in Fig. S4, where global, annual-average percent increase in N10 due to biomass burning decreases from 6.0% without sub-grid





coagulation to 3.1% with sub-grid coagulation and the increase in N80 due to biomass burning decreases from 10.4% to 8.0%. Table 2 shows an overview of how globally, annually averaged percent increase in N10 and N80 due to biomass burning changes between simulations.

**Table 2: Global, annual-mean percent change due to biomass burning in the surface-level N10 and N80 and absolute changes in DRE and AIE due to biomass burning.**

| Compared Simulations | N10 (%) | N80 (%) | externally-mixed DRE (mW m⁻²) | core-shell DRE (mW m⁻²) | AIE (mW m⁻²) |
|---|---|---|---|---|---|
| *noSubCoag- noBB* | 5.9 | 14.5 | -224 | -188 | -76 |
| *SubCoag - noBB* | 2.3 | 9.2 | -231 | -197 | -43 |
| *D150_noSubCoag - noBB* | 1.2 | 4.8 | -222 | -182 | -29 |
| *D150_SubCoag - noBB* | 0.2 | 3.6 | -253 | -214 | -16 |
| *s1.6_noSubCoag - noBB* | 19.4 | 43.0 | -169 | -145 | -155 |
| *s1.6_SubCoag - noBB* | 4.6 | 15.2 | -206 | -177 | -66 |
| *noSubCoag_FINN - noBB* | 6.0 | 10.4 | -214 | -93 | -63 |
| *SubCoag_FINN - noBB* | 3.1 | 8.0 | -128 | -100 | -52 |
| *D150_noSubCoag_FINN - noBB* | 2.0 | 3.4 | -125 | -91 | -31 |
| *D150_SubCoag_FINN - noBB* | 1.1 | 3.2 | -146 | -113 | -26 |
| *s1.6_noSubCoag_FINN - noBB* | 18.3 | 32.0 | -82 | -63 | -112 |
| *s1.6_SubCoag_FINN - noBB* | 5.3 | 13.4 | -105 | -84 | -73 |

Figure 3 shows the annual-mean median diameter and modal width for biomass burning emissions that have been processed with Eqs. 1 and 2 for 24 hours offline. There is large spatial variability in the annual-mean effects of the sub-grid coagulation parameterization on the number median diameter and lognormal modal width. The increase in median diameter and decrease

10 in modal width due to sub-grid coagulation is larger over the Amazon, the Congo, southeast Asia, and parts of the boreal forests in Siberia and North America -- the same regions with the largest biomass burning emissions. These regions have



larger fire areas, lower wind speed, or lower mixing depth on average compared to regions with less biomass burning emissions, which increases the diameter of sub-grid coagulation-processed emissions.

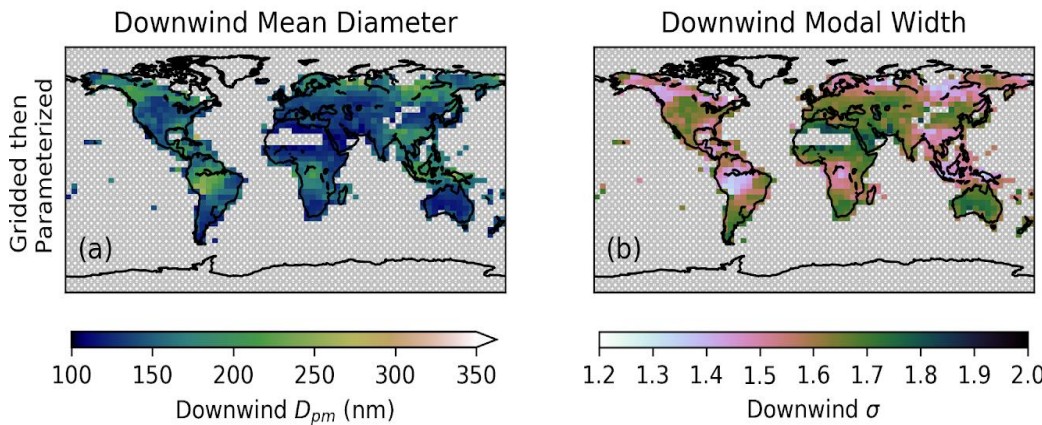

**Figure 3: Annual-mean median diameter (a) and modal width (b) for sub-grid-processed biomass burning emissions predicted for 2010 using the Sakamoto et al. (2016) parameterization after 24 hours of sub-grid coagulation with an emitted median diameter of 100 nm and an emitted modal width of 2. Fire (FINNv1.5) and meteorological data is averaged over a 4°x5° grid and then that gridded data is run through the Sakamoto et al. (2016) parameterization. The regions with grey cross-hatching are grid-cells with no fire data.**

The assumed size distribution of fresh emissions also impacts the simulated percent increase in N80 due to biomass burning (Fig. 4). A larger emission median diameter produces fewer particles initially, leading to a decreased rate of coagulation and thus fewer particles are lost by coagulation. Panels a and b of Fig. 4 show that simulations with the increased emitted median diameter of 150 nm (*D150_noSubCoag* and *D150_SubCoag*, respectively) result in a smaller relative increase in N80 in regions with biomass burning emissions than simulations with the original emitted median diameter of 100 nm (*noSubCoag* and *SubCoag*, panels c and d). This smaller increase in N80 with increasing emitted median diameter is due to roughly 3.4 times fewer particles being emitted when the median diameter is increased by 50% because the same volume must be distributed to larger (spherical) particles. Without sub-grid coagulation (Fig. 4a), this increase in emission diameter leads to a decrease in the biomass burning contribution to N80 in biomass burning regions (particularly in the Amazon). This decreased biomass burning contribution to N80 leads to a reduced global-, annual-average percent increase in N80 due to biomass burning from 14.5% with an emitted median diameter of 100 nm to 4.8% with an emitted median diameter of 150 nm. Including sub-grid coagulation dampens the sensitivity of the biomass burning contribution to N80 to the initial




emission median diameter (sub-grid coagulation has less of an effect when emissions are larger). The same dampening of changes due to sub-grid coagulation can be seen for FINN in Fig. S5.

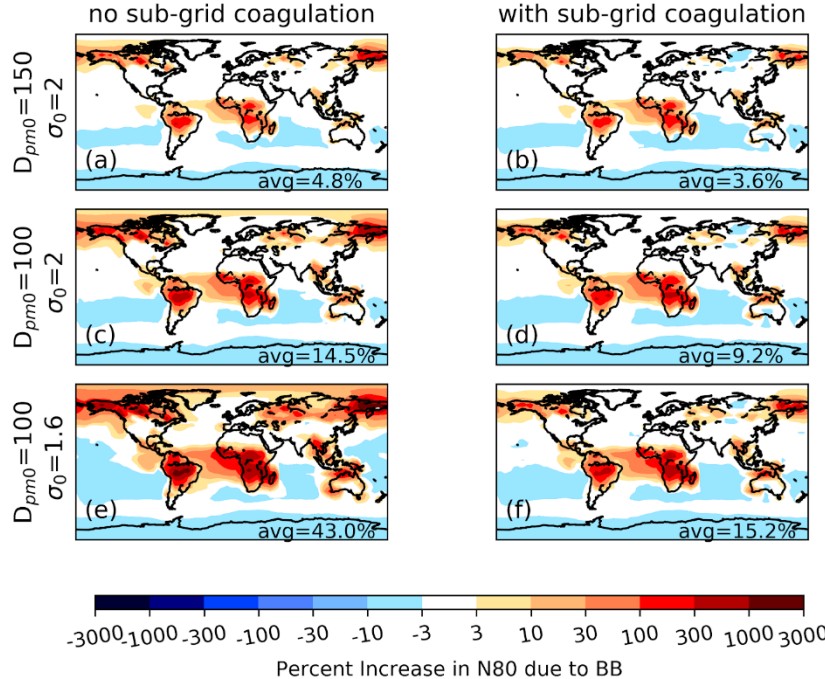

**Figure 4: Annual-average percent changes in N80 at the surface level due to the inclusion of GFED biomass burning emissions relative to the simulation without biomass burning (*noBB*). Panels (a), (c), and (e) have no sub-grid coagulation (*D150_noSubCoag*, *noSubCoag*, and *s1.6_noSubCoag*, respectively). Panels (b), (d), and (f) have sub-grid coagulation (*D150_SubCoag*, *SubCoag*, and *s1.6_SubCoag*, respectively). Panels (a) and (b) have an emitted median diameter of 150 nm and an emitted modal width of 2. Panels (c) and (d) have an emitted median diameter of 100 nm and an emitted modal width of 2. Panels (e) and (f) have an emitted median diameter of 100 nm and an emitted modal width of 1.6. The number in the bottom right of each panel is the global mean percent increase in N80 due to biomass burning.**

Figures 4e and 4f show the increase in N80 due to biomass burning when the emitted modal width is reduced to 1.6 (*s1.6_noSubCoag* and *s1.6_SubCoag*, respectively). This smaller modal width leads to a higher percent increase in N80 because the median diameter is above 80 nm (a higher fraction of the fresh biomass burning particles are larger than 80 nm). Without sub-grid coagulation, this decrease in modal width leads to an increase in the globally, annually averaged percent increase in N80 due to biomass burning from 14.5% with an emitted modal width of 2 to 43.0% with an emitted modal width of 1.6. Similar to what was shown with emitted median diameter in Fig. 4a and b, including sub-grid coagulation results in a smaller change in the biomass burning contribution to N80 relative to the *noSubCoag* assumption because the higher number





of particles increases the rate of coagulation, dampening the effect. With coagulation, the globally, annually averaged increase in N80 due to biomass burning is 9.2% when the emitted modal width is 2 and 15.2% when the emitted modal width is 1.6. Similar responses to changing the emitted modal width are seen for FINN emissions in Fig. S5. Thus, sub-grid coagulation tends to dampen the sensitivity in N80 (CCN-sized particles) due to uncertainty in emission size distribution

parameters in biomass burning plumes.

To further explore the regional effect of coagulation on ambient size distributions, Fig. 5 shows the full ambient size distributions for all simulations with GFED emissions for two biomass burning regions. The June-July-August-mean size distributions over Alaska are shown in panels a and c, and the August-September-October-mean size distributions over the

Amazon are shown in panels b and d. In these two regions, the simulated size distribution is very sensitive to the initial size distribution and whether or not there is sub-grid coagulation. Without biomass burning (*noBB*), Alaska has no nucleation-mode particles. When biomass burning emissions from GFED are included (*noSubCoag*), there is a nucleation mode (due to $SO_2$ emitted by the fires). As shown in Fig. 5, including sub-grid coagulation (*noSubCoag* to *SubCoag*, see Table 1) increases the peak diameter in the accumulation mode from 89 nm to 224 nm and decreases the modal width. The same can

be seen for FINNv1 emissions in Fig. S6.

Figure 5a shows that in Alaska when the emitted median diameter is increased (*noSubCoag* to *D150_noSubCoag*), the ambient peak diameter in the accumulation mode increases from 89 nm to 141 nm, and the number of particles in the accumulation mode decreases. Increasing the emitted median diameter of the case with sub-grid coagulation (*SubCoag* to

*D150_SubCoag*) also increases the ambient peak diameter, but this increase is smaller than the difference between the *noSubCoag* and *D150_noSubCoag* simulations. Panel b shows that similar results are seen for the Amazon.



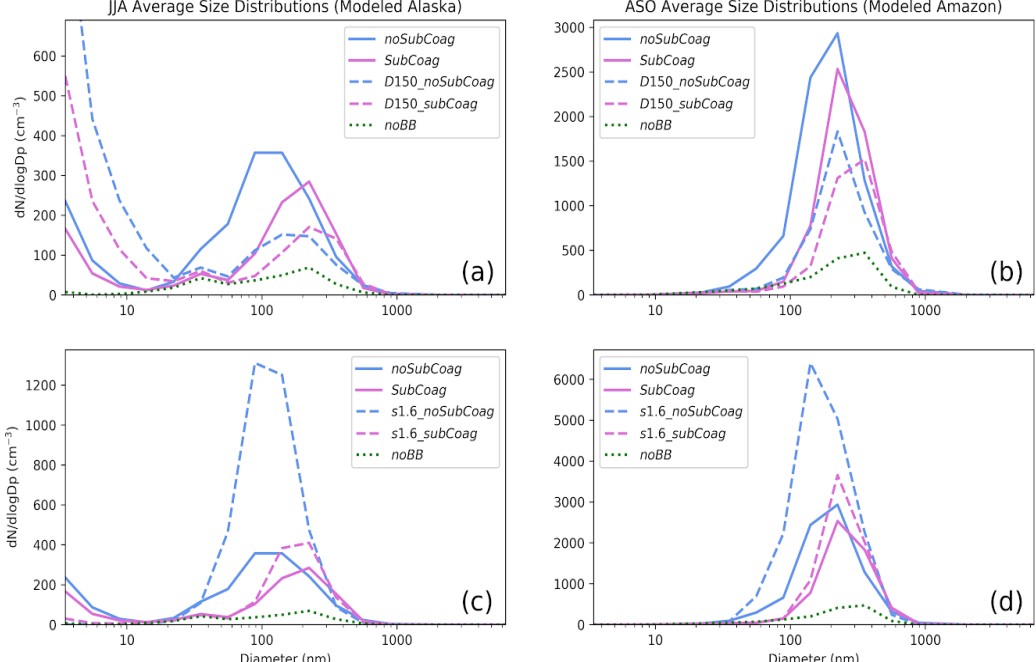

**Figure 5: Predicted, grid-resolved surface-level aerosol size distributions with GFED biomass burning emissions over Alaska at 62° N, 140° W, averaged over the June, July, and August fire season (a and c) and the Amazon at 6° S, 60° E, averaged over the August, September, and October fire season (b and d). All panels show the size distributions for the *noBB*, *noSubCoag*, and *SubCoag* simulations in the dashed green, solid blue, and solid pink lines, respectively. The top panels (a and b) show the sensitivity to the emitted median diameter, and the bottom panels (c and d) show the sensitivity to the emitted modal width. Note the different y-axis scales.**

Figure 5c shows that in Alaska when the initial modal width is increased (*noSubCoag* to *s1.6_noSubCoag*), the number of particles in the accumulation mode increases because the mass emissions remain the same. When the initial modal width is decreased in the case with sub-grid coagulation (*SubCoag* to *s1.6_SubCoag*), the number of particles in the accumulation mode also increases, but this change is smaller than without sub-grid coagulation. Panel d shows similar results for the Amazon. Overall, sub-grid coagulation causes a loss in number at small diameters and a smaller increase in number at the larger diameters in the distribution. This leads to an increase in median (and peak) diameter and a decrease in modal width in the ambient size distribution. Figure S6 shows that these effects of sub-grid coagulation are also present when FINNv1





emissions are used. When sub-grid coagulation is included, the simulated ambient size distribution is less sensitive to the choice of emitted size distribution.

### 3.3 Sensitivity of radiative effects

Figure 6 summarizes our radiative-effect findings with global, annual-average values for each simulation. Table 2 shows the globally, annually averaged biomass burning radiative effects for each simulation. Figures 7, 8, S7, and S8 show our simulated direct radiative effect (DRE) due to biomass burning under assumptions of external mixing (Figs. 7 and S7) and internal mixing (Figs. 8 and S8), and with GFED emissions (Figs. 7 and 8) and FINN emissions (Figs. S7 and S8). The DRE due to biomass burning is sensitive to the initial size distribution, the assumed mixing state of BC, the biomass burning emission inventory, and whether or not there is sub-grid coagulation. As can be seen in Figs. 7 and 8, the DRE due to biomass burning is generally negative (a cooling tendency), but there are regions of slight (up to 660 mW m$^{-2}$ locally) positive DRE over bright surfaces, such as the ice sheets over Greenland and Antarctica, where the biomass burning smoke plume has a lower albedo than the surface. With no sub-grid coagulation, changing the initial median diameter (panels a and c) has little effect, regardless of assumed mixing state. With sub-grid coagulation, increasing emitted median diameter (panel d to b) increases the DRE for both mixing state assumptions. Regardless of sub-grid coagulation, decreasing the initial modal width (panel c to e) decreases the magnitude of the effect globally.





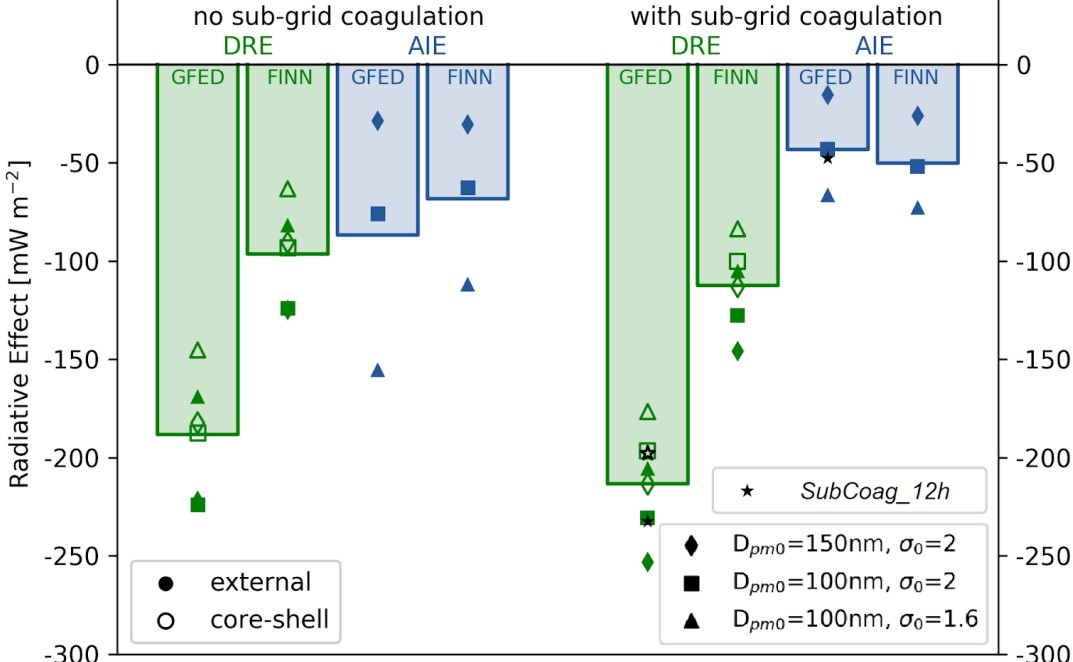

**Figure 6: Global-mean all-sky direct radiative effect due to biomass burning (DRE) and cloud-albedo aerosol indirect effect due to biomass burning (AIE) for all sensitivity simulations with and without sub-grid coagulation. The green bars represent the DRE averaged over all size-distribution, mixing-state sensitivity, and coagulation time sensitivity simulations. The blue bars represent the AIE averaged over all size-distribution and coagulation time sensitivity simulations. The left-hand bars represent simulations using GFED fire emissions and the right-hand bars represent simulations using FINNv1 fire emissions. The diamond, square, and triangle symbol shapes represent the globally averaged value for the different emitted size distributions, as indicated in the legend, with a coagulation time of 24 hours. The star symbol shape represents the globally averaged value for the *SubCoag_12h* case, which has an emitted median diameter of 100 nm and an emitted modal width of 2, like the square symbol case, but with the time spent undergoing sub-grid coagulation reduced from 24 hours to 12 hours, run only with GFED fire emissions. The filled symbols for DRE represent cases with an external mixture and the open symbols represent cases with a core-shell mixture.**

The reason for the greater cooling tendency in DRE with larger particle sizes (either through sub-grid coagulation or larger emissions) is due to the mass scattering and absorption efficiencies (the scattering and absorption cross sections per unit mass). For a refractive index generally representative of biomass burning smoke (1.53 - 0.1$i$; Mack et al., 2010), the diameter of peak mass scattering efficiency is about 400 nm, whereas the mass absorption efficiency is relatively constant across



submicron diameters (Seinfeld and Pandis, 2016). In Janhäll et al. (2010), the average fresh plume has a mass median diameter of 272 nm. In these biomass burning plumes, where the particles tend to have diameters smaller than the diameter of peak mass scattering efficiency, aging the biomass burning particles via coagulation increases the mass scattering efficiency as it increases the diameter of the particles. On the other hand, the mass absorption efficiency changes by little

5    through coagulational aging. Hence, simulations with sub-grid coagulation or larger fresh biomass burning emissions have more-negative DRE values. Finally, decreasing the modal width concentrates the particles around the median diameter, away from the peak of mass scattering efficiency (and with little change to the mass absorption efficiency), decreasing the biomass burning DRE.

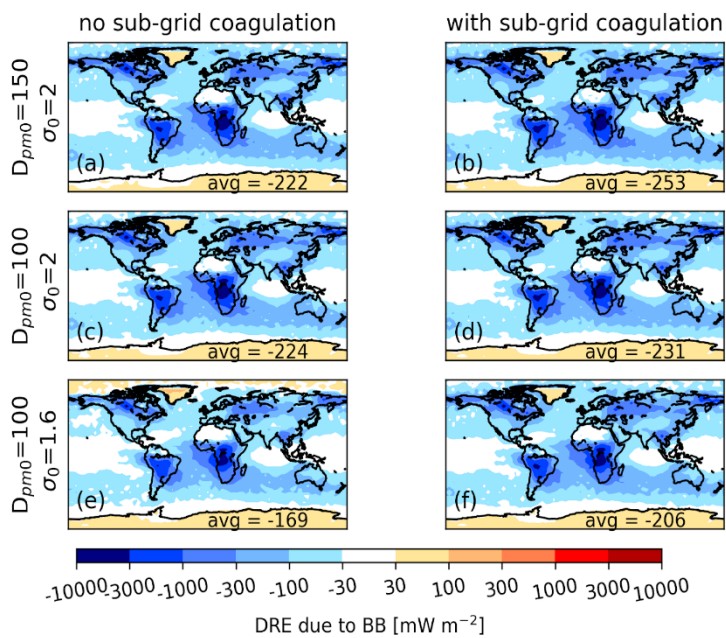

**Figure 7: All-sky direct radiative effect (DRE) due to biomass burning aerosols using GFED emissions and the external-mixing assumption. Panels (a), (c), and (e) are without sub-grid coagulation (*D150_noSubCoag*, *noSubCoag*, and *s1.6_noSubCoag*, respectively). Panels (b), (d), and (f) are with sub-grid coagulation (*D150_SubCoag*, *SubCoag*, and *s1.6_SubCoag*, respectively). Panels (a) and (b) have an emitted median diameter of 150 nm and an emitted modal width of 2. Panels (c) and (d) have an emitted**

15    **median diameter of 100 nm and an emitted modal width of 2. Panels (e) and (f) have an emitted median diameter of 100 nm and an emitted modal width of 1.6. The number in the bottom right of each panel is the global mean DRE value [mW m$^{-2}$].**

Assuming internally mixed BC with a core-shell morphology (Figs. 8 and S8) rather than a fully external mixture (Figs.7 and S7) also decreases the magnitude of the cooling tendency of the DRE in the global mean, as a core-shell morphology





increases absorption. Cases using GFED (Figs. 7 and 8) have a globally, annually averaged DRE that is of much greater magnitude than the cases using FINNv1 (Figs. S7 and S8) because GFED has a larger mass of biomass burning emissions.

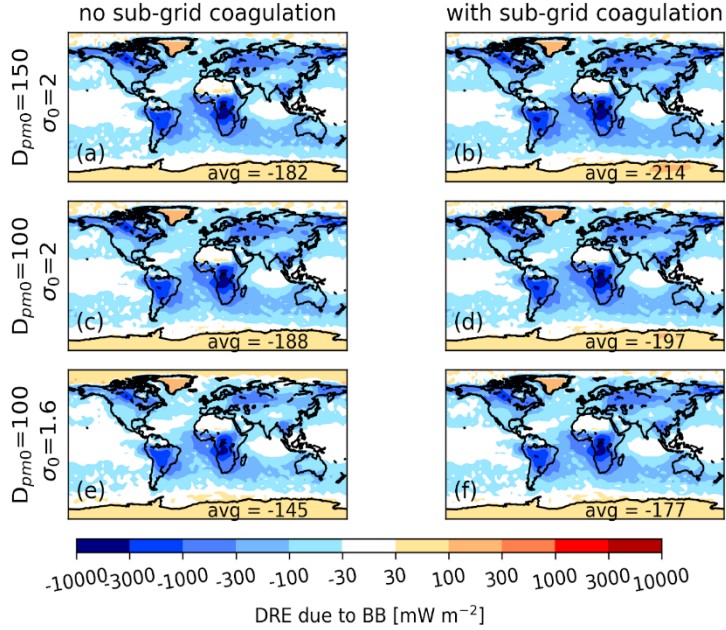

**Figure 8: All-sky direct radiative effect (DRE) due to biomass burning aerosols using GFED emissions and the internal, core-shell mixing assumption. Panels (a), (c), and (e) are without sub-grid coagulation (*D150_noSubCoag*, *noSubCoag*, and *s1.6_noSubCoag*, respectively). Panels (b), (d), and (f) are with sub-grid coagulation (*D150_SubCoag*, *SubCoag*, and *s1.6_SubCoag*, respectively). Panels (a) and (b) have an emitted median diameter of 150 nm and an emitted modal width of 2. Panels (c) and (d) have an emitted median diameter of 100 nm and an emitted modal width of 2. Panels (e) and (f) have an emitted median diameter of 100 nm and an emitted modal width of 1.6. The number in the bottom right of each panel is the global mean DRE value [mW m$^{-2}$].**

Figures 9 and S9 show the biomass burning cloud-albedo aerosol indirect effect (AIE) for GFED and FINNv1, respectively. For all simulations, biomass burning leads to a negative AIE over and downwind of biomass burning regions, and a slight positive AIE (up to 850 mW m$^{-2}$) in many remote regions (due to feedbacks in aerosol nucleation/growth described in Sect. 3.2 in reference to N10 and N80 changes). The strongest cooling is confined to areas where there is both an increase in aerosol number concentration and an environment susceptible to changes in cloud properties, such as areas where there is a low number concentration of CCN and abundant warm clouds. Comparing Fig. 9 to Fig. 4, the spatial distribution in AIE is roughly similar with the inverse changes in N80. The same can be said for FINNv1 by comparing Fig. S9 and Fig. S5. Decreasing the initial modal width, as seen in panels c to e and d to f in the four figures (4, 9, S5, and S9) leads to a 131%


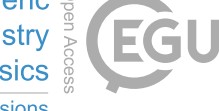


increase on average in the biomass burning contribution to the globally, annually averaged N80 concentrations and therefore a 79% increase in the magnitude of the globally, annually averaged AIE due to biomass burning. A larger initial median diameter, as seen when moving from panel c to a and d to b in the same four figures, leads to a 64% decrease in globally, annually averaged N80 concentrations and a 62% decrease in the magnitude of the globally, annually averaged AIE. Sub-

grid coagulation similarly decreases N80 and therefore decreases the magnitude of the AIE (on average between the three cases) by 49% globally, annually averaged relative to the simulations without sub-grid coagulation. The choice of emissions inventory used has only a small effect, as can be seen in the difference between Fig. 9 and Fig. S9. Lower emissions from FINNv1 lead to less cooling, particularly in the Arctic, but also less warming in remote regions (e.g. the southern hemisphere). These two effects balance out the difference between the emission inventories. When sub-grid coagulation is

used, the AIE is less sensitive to the assumed initial size distribution. This is because sub-grid coagulation acts as a negative feedback on changing the initial size distribution.

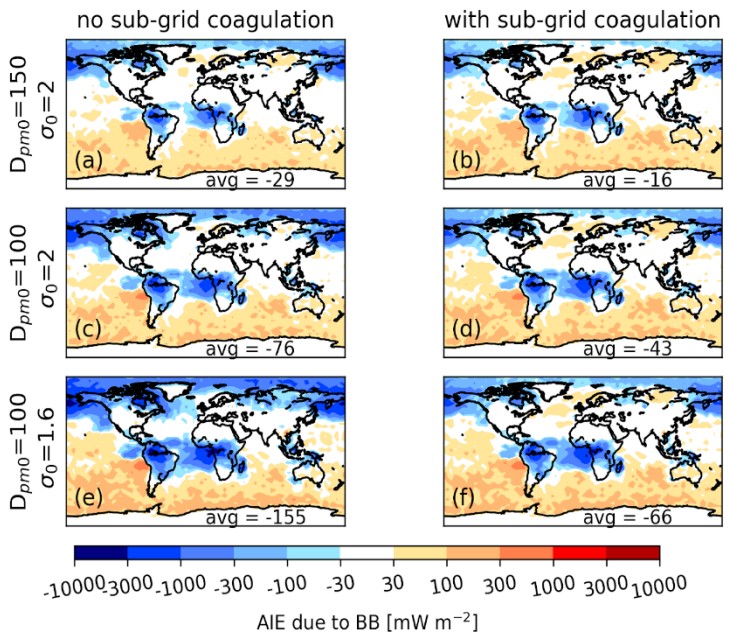

**Figure 9: Cloud-albedo aerosol indirect effect (AIE) due to biomass burning aerosols using GFED emissions. Panels (a), (c), and (e)**

**are without sub-grid coagulation (*D150_noSubCoag*, *noSubCoag*, and *s1.6_noSubCoag*, respectively). Panels (b), (d), and (f) are with sub-grid coagulation (*D150_SubCoag*, *SubCoag*, and *s1.6_SubCoag*, respectively). Panels (a) and (b) have an emitted median diameter of 150 nm and an emitted modal width of 2. Panels (c) and (d) have an emitted median diameter of 100 nm and an emitted modal width of 2. Panels (e) and (f) have an emitted median diameter of 100 nm and an emitted modal width of 1.6. The number in the bottom right of each panel is the global mean AIE value [mW m⁻²].**



As introduced earlier, Fig. 6 summarizes the global biomass burning DRE and AIE estimates from our suite of simulations. The DRE is generally larger in magnitude than the AIE and tends to increase in magnitude with sub-grid coagulation added due to a shift in diameter to a higher mass scattering efficiency. The estimated DRE varies depending on the size

distribution, the mixing state assumed, and the mass of emissions. The magnitude of the AIE globally is smaller with sub-grid coagulation because coagulation reduces the number of particles, thereby reducing the number of CCN from biomass burning. With sub-grid coagulation, the range in AIE due to changes in initial size distribution is much smaller because the coagulation acts as a negative feedback on changes to the initial size distribution.

## 3.4 Limitations of this study

While we have shown that the results from this study capture the changes that occur when sub-grid coagulation is included in the model, there are limitations to our analysis:

- In this study, we did not compare our results to measured ambient size distributions in smoke-impacted regions. There are limited observations of long-term ambient aerosol size distributions in the regions where this study finds sub-grid coagulation to be impactful. The GoAmazon field campaign (Martin et al., 2016) fits our needs, but the

measurement site was located near the city of Manaus, leading to heterogeneity at scales much smaller than the model can resolve (U.S. Department of Energy, 2014). The impacts of the urban plume change the observations in ways that we cannot quantify and is beyond the scope of this study. Weigum et al. (2016) describes the challenges of comparing point measurements to coarse models. As such, comparison between model and measurements is left for future work. Measurements directly of biomass burning plumes (e.g., from field campaigns) would also not be

representative of the grid-scale mean size distributions represented by the model.

- In the current model set-up, it is assumed that smoke plumes do not overlap. Overlapping of smoke plumes would lead to a higher initial number concentration and therefore more sub-grid coagulation. The impact on N80 when the smoke plumes are allowed to completely overlap (i.e., all fires in the gridbox form one "superplume") is shown in Fig. S10. Because sub-grid coagulation is enhanced, there is a reduction in the impact of biomass burning on N10

and N80 when smoke plumes overlap completely instead of not overlapping at all.

- This study assumes that smoke plumes are emitted within the boundary layer, as is done in GEOS-Chem. This allows the plume mixing depth to be the same as the boundary layer depth. Rémy et al, (2003) found that most plumes do fit this category, but further work may allow for emissions higher than the boundary layer, and in these cases, the mixing depth may be more challenging to quantify (Zhu et al., 2018).

- The emitted size distribution from fires varies depending on fire characteristics (Janhäll et al., 2010; Levin et al., 2010), while in this study the same emitted size distribution is applied to all fires. Ideally, fresh size distributions would be linked to fire characteristics in future emission inventories.



- It is assumed in this study that all fires in the same gridbox on a given day are the same size. Fire size can have an impact on sub-grid coagulation (Hodshire et al, 2018). If fires vary greatly in size within a gridbox (at a specific time), larger fires may have more sub-grid coagulation effects than small fires in the same gridbox. Figure S2 shows that using the parameterization on each individual fire, accounting for size, and then averaging the results gives approximately the same downwind median diameter and modal width as using the parameterization on the gridded fires with the assumption that all fires in the same gridbox are the same size.

- In this study it is assumed, as described in the methods, that the smoke plume spends 24 hours aging. While this timescale is potentially variable for each fire, the exponent on this variable in the parameterization equations (Eqs. 1 and 2) is significantly below 1 and so the effect is dampened. We performed a sensitivity study with an aging time of 12 hours (*SubCoag_12h*), which is otherwise the same as the *SubCoag* case. Figure S11 shows that the annual-average change in N80 due to biomass burning emissions depends more on whether there is sub-grid coagulation included than on whether the sub-grid coagulation timescale is 12 hours or 24 hours. Similar results are shown in Fig. S12 for the direct radiative effect due to biomass burning (DRE) using an external mixing assumption, in Fig. S13 for the DRE using an internal, core-shell mixing assumption, and in Fig. S14 for the cloud-albedo aerosol indirect effect due to biomass burning. Figure 6 shows that the global, annual-average biomass burning aerosol radiative effects for the *SubCoag_12h* case are nearly the same as for the *SubCoag* case.

- This study only investigated the effects of including sub-grid coagulation, but other sub-grid processes are occuring in biomass burning plumes. Organic aerosol in biomass burning plumes can evaporate and can be formed through condensation. The rates of this evaporation and condensation may have dependencies on fire size and dilution, similar to coagulation here, and may be more important than coagulation in small plumes (Bian et al., 2017; Hodshire et al., 2018). These processes are not considered in this model.

- Assumptions about mixing state were made when calculating the DRE and AIE due to lack of explicit simulation of the mixing state and limited knowledge of the mixing state of biomass burning emissions.

- When calculating the aerosol radiative effects, monthly mean aerosol concentrations and meteorological inputs were used to increase computation efficiency.

## 4 Conclusions

In this paper, we use a global chemical-transport model with aerosol microphysics and a parameterization of sub-grid biomass burning coagulation to estimate the impacts of sub-grid coagulation on the ambient size distribution and aerosol radiative effects. Including sub-grid coagulation (moving from the *noSubCoag* simulation to the *SubCoag* simulation in Table 1) decreases the magnitude of the biomass burning global, annual-mean cloud-albedo aerosol indirect effect (AIE) by 43% from -76 mW m$^{-2}$ to -43 mW m$^{-2}$, as it reduces the number concentration of CCN-sized particles from biomass burning emissions. Sub-grid coagulation increases the biomass burning global-, annual-mean direct radiative effect (DRE) by 4%



from -206 mW m$^{-2}$ to -214 mW m$^{-2}$ (on average between the external mixing assumption and the internal, core-shell mixing assumption) due to an increase in mass scattering efficiency. However, different assumptions in initial size distribution, emission inventory, and particle morphology can also have a large effect on the magnitude of the AIE and DRE and the effect of coagulation.

We test a series of sensitivities to account for uncertainties in the effect of sub-grid coagulation on particle size distributions in the smoke plume: varying the initial median diameter and modal width, using two biomass burning emissions inventories (GFED and FINNv1), varying the time spent undergoing sub-grid coagulation, and testing the DRE under two mixing states (external and internal core-shell). Testing these sensitivities, global, annual-average AIE due to biomass burning ranges from

10    -29 to -155 mW m$^{-2}$ without sub-grid coagulation. With sub-grid coagulation, the absolute magnitude and range of the globally, annually averaged AIE due to biomass burning reduces such that it ranges from -16 to -66 mW m$^{-2}$. This range is reduced due to sub-grid coagulation homogenizing the number of particles generated by biomass burning. Emissions with a smaller emitted median diameter have a greater number concentration, all else being equal, which leads to more coagulation, reducing number concentration and increasing the median diameter. Emissions with a larger modal width have an increased

rate of coagulation, which reduces the modal width. Through this homogenizing effect of sub-grid coagulation, changes to the emitted size distribution have less effect on the final size distribution when sub-grid coagulation is included than they would without sub-grid coagulation. Regardless of initial size distribution or emission inventory, the inclusion of sub-grid coagulation decreases the global AIE magnitude.

The DRE due to biomass burning ranges from -145 to -224 mW m$^{-2}$ without sub-grid coagulation and from -177 to -253 mW m$^{-2}$ with sub-grid coagulation. This range of values comes from difference in the size distribution of the particles, the mass of emissions between the two inventories, and the assumed mixing state. Most of the uncertainty in DRE is due to the emission inventory selection. GFED generally has 95% more cooling (averaged between all sensitivity cases) due to annually, globally averaged DRE than FINN. This difference in DRE is because GFED has higher mass emissions, which

increases DRE magnitude, and a higher OA to BC ratio, which increases cooling. Regardless of these assumptions, we find that sub-grid coagulation increases the magnitude of the estimated DRE by moving the aerosol size distribution into sizes more efficient at scattering.

Regarding the limitations of this study discussed in Section 3.4, we have several recommendations for future work. We did

not compare our results to measured ambient size distributions in smoke-impacted regions in this work and this should be done in future work, acknowledging the changes described in the limitations. Further, it would be useful to determine a more accurate value for the mixing depth of smoke plumes. In this paper, boundary layer height was used but depending on the injection depth of the plume that may or may not be realistic. Finally, OA in biomass burning plumes undergoes evaporation and SOA formation, and these rates may also depend on fire size and dilution, similar to coagulation here (Bian et al., 2017;




Hodshire et al., 2018). Future work should focus on parameterizing these sub-grid OA effects similar to the coagulation parameterization of Sakamoto et al. (2017).

**Author contributions:**

ER, JRP, CRL, and MJA defined the scientific questions and scope of this work. ER performed all GEOS-Chem model simulations and radiative forcing calculations with help from JKK and JRP. ER prepared the primary text with substantial contributions from all authors.

**Competing interests:**

The authors declare that they have no conflict of interest.

**Acknowledgements:**

This research was supported by the U.S. National Science Foundation, Atmospheric Chemistry Program, under Grant No. AGS-1559607 and the U.S National Oceanic and Atmospheric Administration, an Office of Science, Office of Atmospheric Chemistry, Carbon Cycle, and Climate Program, under the cooperative agreement award #NA17OAR430001.

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
