# Peer review of "Effects of Near-Source Coagulation of Biomass Burning Aerosols on Global Predictions of Aerosol Size Distributions and Implications for Aerosol Radiative Effects"

_Atmospheric Chemistry and Physics, 2018_

## Referee Comment (RC1) · Anonymous Referee #1 · 11 Jan 2019

This study used a sub-grid coagulation parameterization for biomass burning plumes in the GEOS-Chem-TOMAS global aerosol microphysical model and showed large impacts of biomass burning sub-grid coagulation on aerosol number concentrations, aerosol size distributions, and aerosol direct and first indirect effects. The authors found sub-grid coagulation reduced the impact of biomass burning aerosols on number concentrations of particles larger than 80 nm by 37% globally and that this reduction changed estimates of aerosol direct and first indirect effects of biomass burning aerosols by 4% (from -206 mW m-2 to -214 mW m-2) and by 43% (from -76 mW m-2

to -43 mW m-2), respectively. The authors demonstrated that the inclusion of biomass burning sub-grid coagulation significantly reduced the sensitivity of aerosol number concentrations, CCN concentrations, and aerosol-cloud interactions to the treatment of aerosol size distributions at emissions.

The topic of this work is interesting and well suited to the scope of Atmospheric Chemistry and Physics. The manuscript is well written and the findings by the authors will be useful for estimating aerosol-climate interactions more accurately. Overall, the manuscript should be accepted by this journal after minor revisions. Some minor comments, which should be addressed before acceptance, are described below.

Minor comments:

1. Page 1, Lines 23-24:

external mixing –> external mixing of black carbon

internal mixing –> internal mixing of black carbon

2. Page 2, Lines 20-21:

Please add the following reference: H. Matsui (2016), doi:10.1002/2015JD023998.

3. Page 4, Line 24:

Please add a few sentences on the treatment of SOA formation in the global aerosol model.

4. Page 4, Lines 23-31:

Please clarify gaseous and aerosol species considered in the biomass burning emissions in the author's global model.

5. Page 6, Line 24:

In equation (1), 84.56 should be 84.576, based on Sakamoto et al. (2016).

6. Page 7, Line 16:

In Sakamoto et al. (2016), their parameterization is based on their simulations conducted for 5 hours during biomass burning emissions. This parameterization was extended to 24 and 12 hours in the current study. Can the authors justify this extension?

I suggest the authors to confirm this extension does not overestimate sub-grid coagulation rate (because coagulation rate will be slower with time) and to add some discussions to the text.

7. Page 7, Lines 26-29:

Scatterplots and correlation coefficients may be useful.

8. Page 8, Line 4:

Please add a sentence that the two assumptions of mixing state have the same aerosol number concentrations and size distributions in total.

9. Figure 2 and related figures:

The main focus of this study is the inclusion of sub-grid coagulation. So, I think the difference between SubCoag and noSubCoag is the most important point in this discussion (rather than the differences from noBB). How about adding plots on the difference between SubCoag and noSubCoag (absolute value or percent)? The authors can add similar difference plots (between SubCoag and noSubCoag) to other figures (Figures 4 and 7-9).

10. Figure 6:

This is a nice figure and should be used to summarize conclusions obtained in this study. I suggest the authors to move this figure to the last paragraph of section 3.3 (after Figure 9).

Similar to comment 9, differences between SubCoag and noSubCoag can be added to

this figure. Adding them will make the impact of sub-grid coagulation on DRE and AIE clearer.

11. Page 19, Line 19 – Page 20, Line 6:

Please clarify how the authors estimated the statistic values in this paragraph (131%, 79%, 64%, 62%, and 49%).

12. Page 21, Lines 21-25:

Can the authors add some statistics for more quantitative discussions of this paragraph?

13. Section 3.4:

In addition to the points raised by the authors, I suggest to add the following two points to this section.

Firstly, the simulations made by the authors are for year 2010 only. Biomass burning emissions and meteorological conditions have large year-to-year variability. Please add some discussions on the features of biomass burning emission in 2010 (compared with other years) and on their potential impacts on the estimation of sub-grid coagulation importance.

Secondly, the uncertainty ranges of DRE and AIE in this study (e.g. Figure 6) were estimated from sensitivity simulations with changing single parameter at one time (e.g., median size of emissions, sigma of emissions, mixing state, sugcoag timescale, biomass burning emission data). However, in the real atmospheric conditions, multiple parameters change simultaneously. Therefore, the uncertainty ranges of DRE and AIE in the real atmosphere might be larger than those estimated with single parameter change (conducted in this study). The authors can add discussions on the potential importance of this effect.

14. Page 23, Line 26:

the estimated DRE –> the estimated DRE (increases cooling)

15. Page 24, Line 2:

Sakamoto et al. (2017) –> Sakamoto et al. (2016)

––––––––––––––––––––––––––––––––––

---

## Referee Comment (RC2) · Anonymous Referee #2 · 4 Mar 2019

The paper reports a study of the impacts of coagulation of particles in biomass burning plumes on the climate impacts of biomass burning aerosol. The study finds that this process, that is not usually included in atmospheric or climate models, reduces the number of cloud droplet forming particles produced by biomass burning by 37% globally. Overall, the study finds that including coagulation of particles in biomass burning plumes reduces the cooling impact of biomass burning aerosol through the aerosol indirect effect, but increases the cooling impact through the direct radiative effect.

This is an important study. The paper is well-written. The model experiments are

clearly described and the authors have tested a range of assumptions and datasets. I recommend publication after any minor comments have been addressed.

Minor comments

Section 2.1. What new particle formation scheme and SOA formation did you include in the model? These will control the baseline particle number and the growth rates of particles and so are important for your study.

Figure 6. Do you report the values averaged over all size distributions? From the figure it looks like the impact of coagulation on the average DRE value is greater than the 4% in the Abstract?

Section 2.2 How would your results depend on parameter uncertainty in equations (1) and (2) on Page 6. The authors should be commended for exploring the uncertainty in the global model inputs/datasets. It is probably beyond scope to explore the impact of uncertainty in these equations, but a short discussion would be useful.

Section 2.2 Do you have information on the values of Dpm and model width calculated from equations (1) and (2)? It would be interesting to know the mean values used as input to GEOS-chem as well as spatial and temporal variability.

An important point is to what extent the emitted size distribution in the model represents fresh or aged smoke. This is mentioned by the authors in Section 2.2. Could the treatment of in-plume coagulation simply be captured by assuming a larger emitted size? Or does the in-plume calculations allow treatment of important spatial and temporal variability that would be ignored by using a globally uniform value?

---

## Author Response (AR1)

**RESPONSE TO REFEREE #1:**

This study used a sub-grid coagulation parameterization for biomass burning plumes in the GEOS-Chem-TOMAS global aerosol microphysical model and showed large impacts of biomass burning sub-grid coagulation on aerosol number concentrations, aerosol size distributions, and aerosol direct and first indirect effects. The authors found sub-grid coagulation reduced the impact of biomass burning aerosols on number concentrations of particles larger than 80 nm by 37% globally and that this reduction changed estimates of aerosol direct and first indirect effects of biomass burning aerosols by 4% (from -206 mW m-2 to -214 mW m-2) and by 43% (from -76 mW m-2 to -43 mW m-2), respectively. The authors demonstrated that the inclusion of biomass burning sub-grid coagulation significantly reduced the sensitivity of aerosol number concentrations, CCN concentrations, and aerosol-cloud interactions to the treatment of aerosol size distributions at emissions. The topic of this work is interesting and well suited to the scope of Atmospheric Chemistry and Physics. The manuscript is well written and the findings by the authors will be useful for estimating aerosol-climate interactions more accurately. Overall, the manuscript should be accepted by this journal after minor revisions. Some minor comments, which should be addressed before acceptance, are described below.

*We thank this reviewer for their helpful and thoughtful review. Our responses throughout are in italics.*

Minor comments:

1. Page 1, Lines 23-24:
external mixing –> external mixing of black carbon
internal mixing –> internal mixing of black carbon

*Yes, better to be more specific here. We have made this change.*

2. Page 2, Lines 20-21:
Please add the following reference: H. Matsui (2016), doi:10.1002/2015JD023998.

*We have added this reference.*

3. Page 4, Line 24:
Please add a few sentences on the treatment of SOA formation in the global aerosol model.

*To clarify this, we have made this addition to Section 2.1: "Nucleation rates are parameterized with binary nucleation (Vehkamaki et al., 2002) and ternary nucleation (Napari et al., 2002) scaled globally by a tuning factor of 10-5 (Jung et al., 2010; Westervelt et al., 2013). Secondary organic aerosol includes a 19 Tg yr$^{-1}$ biogenic contribution and a 100 Tg yr$^{-1}$ anthropogenically enhanced contribution correlated with anthropogenic CO emissions (D'Andrea et al., 2013), following the approach of Spracklen et al. (2011)".*

4. Page 4, Lines 23-31:
Please clarify gaseous and aerosol species considered in the biomass burning emissions in the author's global model.

*To clarify this, we have made this addition to Section 2.1: "In our simulations, GFED and FINN biomass burning emissions include nitric oxide, carbon monoxide, sulfur dioxide, ammonia, all alkanes except for methane, acetone, methyl ethyl ketone, formaldehyde, acetaldehyde, alkenes with continuous carbon chains longer than two carbons, black carbon aerosol, and organic aerosol. The FINN biomass burning emissions also include hydroxyacetone and glycoaldehyde."*

5. Page 6, Line 24:

In equation (1), 84.56 should be 84.576, based on Sakamoto et al. (2016).

*We have made this change.*

6. Page 7, Line 16:
In Sakamoto et al. (2016), their parameterization is based on their simulations conducted for 5 hours during biomass burning emissions. This parameterization was extended to 24 and 12 hours in the current study. Can the authors justify this extension?

I suggest the authors to confirm this extension does not overestimate sub-grid coagulation rate (because coagulation rate will be slower with time) and to add some discussions to the text.

*Yes, we do extrapolate on the fits and this should be explicitly discussed. The Sakamoto fits depend on time to a power of less than 0.5; hence, the coagulation does slow with time in the fits. We have made this addition to Section 2.2: "In Sakamoto et al. (2016), simulations used to develop this parameterization are five hours long, so we are extrapolating their fits. Because of the dependence on time to a power less than 0.5 (Eqns. 1 and 2). The impact of coagulation slows with time, which reduces potential errors associated with extrapolating. To test the sensitivity to this 24-hour assumption we include an additional simulation (SubCoag_12h) where conditions are similar to SubCoag but with 12 hours instead of 24 hours of aging."*

7. Page 7, Lines 26-29:
Scatterplots and correlation coefficients may be useful.

*We have added these plots to the supplement and made this addition to the main text as a brief discussion: "Further, Figs. S4 and S5 show that most gridboxes report similar $D_{pm}$ and $\sigma$ using either method, especially at smaller $D_{pm}$ and larger $\sigma$ values, where most gridboxes lie."*

8. Page 8, Line 4:
Please add a sentence that the two assumptions of mixing state have the same aerosol number concentrations and size distributions in total.

*We have made this addition to Section 2.3: "Both mixing states have the same aerosol number concentrations and size distributions in total."*

9. Figure 2 and related figures:
The main focus of this study is the inclusion of sub-grid coagulation. So, I think the difference between SubCoag and noSubCoag is the most important point in this discussion (rather than the differences from noBB). How about adding plots on the difference between SubCoag and noSubCoag (absolute value or percent)? The authors can add similar difference plots (between SubCoag and noSubCoag) to other figures (Figures 4 and 7-9).

*We had debated the idea of making "percent change in the biomass burning impact" plots (i.e. how does the impact of BB on N10 and N80 change by adding sub-grid coagulation of biomass burning aerosol) when writing the paper; however, these plots were challenging to interpret because there regions where N10 and N80 (1) increase due to biomass burning in both the SubCoag and noSubCoag simulations, (2) decrease due to biomass burning in both simulations, and (3) increase in one simulation and decrease in the other. However, to address this reviewer comment, we have made a version of these plots where we require \*both\* the SubCoag and noSubCoag simulations to have at least a 1% increase in N10 and N80 (relative to noBB) in a gridbox, and all other gridboxes are masked. We have added these figures (one for the base emissions assumptions and another for the sensitivity emissions assumptions as Figs. S8 and S10, respectively) to the supplement and mentioned them in the main text. In this figure, the interpretation is straightforward; blue colors show that sub-grid*

*coagulation reduces the impact of biomass burning in that location; red colors show that sub-grid coagulation increases the impact of biomass burning in that location (due to microphysics feedbacks: increased nucleation etc).*

*We also made analogous "percent change in the biomass burning impact" figures that correspond to the three radiative forcing figures in the main text (Figures 7-9) and also added these figures to the supplement (Figs. S13, S15, S17) with reference in the main text.*

10. Figure 6:
This is a nice figure and should be used to summarize conclusions obtained in this study. I suggest the authors to move this figure to the last paragraph of section 3.3 (after Figure 9). Similar to comment 9, differences between SubCoag and noSubCoag can be added to this figure. Adding them will make the impact of sub-grid coagulation on DRE and AIE clearer.

*We have chosen to leave this figure where it currently is because we would like the reader to be able to look at all of the results in context as they read this section, but our intention is for this to be a summary figure with its main discussion coming at the end of the section. We have made this addition to Section 3.3 to clarify: "Figure 6 summarizes our radiative-effect findings with global, annual-average values for each simulation."--> "Figure 6 summarizes our radiative-effect findings with global, annual-average values for each simulation, and we will discuss this figure in detail at the end of this section."*

11. Page 19, Line 19 – Page 20, Line 6:
Please clarify how the authors estimated the statistic values in this paragraph (131%, 79%, 64%, 62%, and 49%).

*Originally, these numbers were used in an attempt to include all sensitivity cases in the calculation. For the numbers comparing two initial size-distribution assumptions, the cases with sub-grid coagulation of biomass burning aerosol and the cases without were both used in the calculation. For the numbers discussing the addition of sub-grid coagulation of biomass burning aerosol, all initial size distribution cases were used in the calculation. This approach made things unnecessarily complicated, and the numbers have been replaced with simpler calculations that should be more intuitive for the reader. For the numbers comparing two initial size distributions, the case without sub-grid coagulation of biomass burning aerosol is now used alone. For the numbers discussing the addition of sub-grid coagulation of biomass burning aerosol, only the default initial size distribution case is now used ($Dpm0$ being 100 nm and $\sigma 0$ being 2).*

*These changes are in section 3.3 where the text is now, "Biomass burning contributes nearly three times more to N80 when decreasing the modal width (a 43% increase in global N80 from adding biomass burning with the decreased width versus a 14.5% increase from the original width) without sub-grid coagulation of biomass burning aerosol, as seen in panels c to e in the four figures (4, 9, S9, and S16). This increase in N80 leads to a 104% increase in the magnitude of the globally, annually averaged AIE due to biomass burning, from -76 mW m-2 to -29 mW m-2. Biomass burning contributes only about one-third as much N80 when increasing the median diameter (a 4.8% increase in global N80 from adding biomass burning with the larger median diameter verses a 14.5% increase from the original median diameter) without sub-grid coagulation of biomass burning aerosol, as seen in panels c to a in the same four figures. This decrease in N80 leads to a 62% decrease in the magnitude of the globally, annually averaged AIE due to biomass burning, from -76 mW m-2 to -29 mW m-2. Sub-grid coagulation similarly decreases the biomass-burning contribution to N80 from 14.5% to 9.2% when assuming the original initial size distribution case ($Dpm0 =100$ nm and $\sigma 0 = 2$). Hence, sub-grid coagulation decreases the biomass-burning AIE by 43% globally (-76 mW m-2 to -43 mW m-2)".*

12. Page 21, Lines 21-25:
Can the authors add some statistics for more quantitative discussions of this paragraph?

*To make this discussion quantitative, the bullet point in question now reads "In the current model set-up, it is assumed that smoke plumes do not overlap. Overlapping of smoke plumes would lead to a higher initial number concentration and slower dilutions and therefore more sub-grid coagulation. The impact on N80 when the smoke plumes are allowed to completely*

*overlap (i.e., all fires in the gridbox form one "superplume") is shown in Fig. S18. Without sub-grid coagulation of biomass burning aerosol, biomass burning increases N80 by 10.4% (globally and annually averaged). With sub-grid coagulation of biomass burning aerosol as it is generally presented in this paper (no overlapping plumes), the increase in global N80 due to biomass burning is decreased by about a quarter to 8.0%. When sub-grid coagulation of biomass burning aerosol includes total overlap of the smoke plumes, the increase in global N80 due to biomass burning is further decreased to only 0.5%. This strong sensitivity of our results to plume overlap highlights that the degree of plume overlap likely needs to be understood".*

13. Section 3.4:
In addition to the points raised by the authors, I suggest to add the following two points to this section.

Firstly, the simulations made by the authors are for year 2010 only. Biomass burning emissions and meteorological conditions have large year-to-year variability. Please add some discussions on the features of biomass burning emission in 2010 (compared with other years) and on their potential impacts on the estimation of sub-grid coagulation importance.

*This is a very good point. We have added the following caveat to Section 3.4: "We only simulated emissions and meteorology for 2010. As there is significant interannual variability in biomass burning emissions as well as the meteorological inputs to the Sakamoto et al. (2016) parameterization, we expect that our results are at least somewhat sensitive to the choice of year (O'Dell et al., 2019)".*

*We did not do a detailed comparison of emissions between years. Given the range of factors that contribute to the impact of biomass burning on particle concentrations, radiative forcing, and the Sakamoto et al. (2016) parameterization, it would be very challenging to speculate about how the results for 2010 may specifically be different from other years.*

Secondly, the uncertainty ranges of DRE and AIE in this study (e.g. Figure 6) were estimated from sensitivity simulations with changing single parameter at one time (e.g., median size of emissions, sigma of emissions, mixing state, subcoag timescale, biomass burning emission data). However, in the real atmospheric conditions, multiple parameters change simultaneously. Therefore, the uncertainty ranges of DRE and AIE in the real atmosphere might be larger than those estimated with single parameter change (conducted in this study). The authors can add discussions on the potential importance of this effect.

*We have made this addition to Section 3.4: "Because sensitivity simulations varied only one parameter at a time, uncertainty ranges of DRE and AIE in the real atmosphere may be larger than those estimated in this study. Uncertainty in other factors -- such as emissions, clouds and their susceptibility, and brown carbon -- also affect DRE and AIE uncertainty ranges".*

14. Page 23, Line 26:
the estimated DRE –> the estimated DRE (increases cooling)

*We have made this change.*

15. Page 24, Line 2:
Sakamoto et al. (2017) –> Sakamoto et al. (2016)

*We have made this change.*

**RESPONSE TO REFEREE #2:**

The paper reports a study of the impacts of coagulation of particles in biomass burning plumes on the climate impacts of biomass burning aerosol. The study finds that this process, that is not usually included in atmospheric or climate models, reduces the number of cloud droplet forming particles produced by biomass burning by 37% globally. Overall, the study finds that including coagulation of particles in biomass burning plumes reduces the cooling impact of biomass burning aerosol through the aerosol indirect effect, but increases the cooling impact through the direct radiative effect.

This is an important study. The paper is well-written. The model experiments are clearly described and the authors have tested a range of assumptions and datasets. I recommend publication after any minor comments have been addressed.

*We thank this reviewer for their helpful and thoughtful review. In particular, we are grateful for this reviewer agreeing to review the paper late in the open discussion and providing comments quickly. Our responses throughout are in italics.*

Minor comments:

Section 2.1. What new particle formation scheme and SOA formation did you include in the model? These will control the baseline particle number and the growth rates of particles and so are important for your study.

*To clarify this, we have made this addition to Section 2.1: "Nucleation rates are parameterized with binary nucleation (Vehkamaki et al., 2002) and ternary nucleation (Napari et al., 2002) scaled globally by a tuning factor of 10-5 (Jung et al., 2010; Westervelt et al., 2013). Secondary organic aerosol includes a 19 Tg yr−1 biogenic contribution and a 100 Tg yr−1 anthropogenically enhanced contribution correlated with anthropogenic CO emissions (D'Andrea et al., 2013), following the approach of Spracklen et al. (2011)."*

Figure 6. Do you report the values averaged over all size distributions? From the figure it looks like the impact of coagulation on the average DRE value is greater than the 4% in the Abstract?

*This is a very good point. The 4% value does not adequately capture the effect that sub-grid coagulation of biomass burning aerosol has on the average DRE. The values listed in the conclusions, which is where the 4% value is reported, were only for the default initial size distribution case where Dpm0 is 100 nm and σ0 is 2. To clarify this, we added the following sentence, "Sub-grid coagulation increases the biomass burning global-, annual-mean direct radiative effect (DRE) by 4% from -206 mW m-2 to -214 mW m-2 due to an increase in mass scattering efficiency for the default initial size distribution with an initial median diameter of 100 nm and an initial lognormal modal width of 2 (on average between the external mixing assumption and the internal, core-shell mixing assumption)".*

*In Fig. 6, the DRE is not affected very much by the inclusion of sub-grid coagulation of biomass burning aerosol when only the default case (filled square) is considered. However, the sensitivity cases can vary much more. To include the other initial size distribution sensitivity cases, we have added the following to the conclusions: "In our sensitivity cases testing different initial size distributions, described below, the DRE is more affected by the presence or absence of sub-grid coagulation of biomass burning aerosol, changing as much as 22%".*

Section 2.2 How would your results depend on parameter uncertainty in equations (1) and (2) on Page 6. The authors should be commended for exploring the uncertainty in the global model inputs/datasets. It is probably beyond scope to explore the impact of uncertainty in these equations, but a short discussion would be useful.

*This is a good thing to discuss explicitly. We have added the following text to Section 2.2: "In Sakamoto et al., (2016), these equations (Eqns. 1 and 2 here) explain 77-79% of the variability in Dpm and in their plume simulations. Hence, there are*

*uncertainties in our analyses introduced by the simple form of Eqns. 1 and 2; however, we expect these uncertainties to be smaller than the uncertainties in biomass burning emission inventories, plume overlap, fire size, mixing height, etc.".*

Section 2.2 Do you have information on the values of Dpm and model width calculated from equations (1) and (2)? It would be interesting to know the mean values used as input to GEOS-chem as well as spatial and temporal variability.

*The spatial variability of the annual-mean values for these two variables was given in Figure 3, though this was not introduced until Section 3.2. We have moved its introduction to Section 2.2 (now becoming Figure 1) with further discussion remaining in Section 3.2.*

*To show temporal variability, we have now plotted the timeseries for the sub-grid-plume processed $D_{pm}$ and $\sigma$ for the two locations investigated in Figure 5 (Alaska and the Amazon) as Fig. S2.. In both of these regions, there is day-to-day variability driven by changes in wind, fire size, and mixing depth. There is also an apparent seasonal cycle in the values at both locations, where particles are larger and have a narrower distribution at the peak of the fire season, which is likely driven by larger fire sizes during these times. We have added these figures to the SI and added a brief reference to them in Section 2.2.*

An important point is to what extent the emitted size distribution in the model represents fresh or aged smoke. This is mentioned by the authors in Section 2.2. Could the treatment of in-plume coagulation simply be captured by assuming a larger emitted size? Or does the in-plume calculations allow treatment of important spatial and temporal variability that would be ignored by using a globally uniform value?

*Given the spatial and temporal variability in the sub-grid-processed size distributions in Figure 1 (previously Figure 3) and Figure S2, it appears that any fixed global assumption of an "aged" biomass burning size distribution may underestimate regional variability. We have added text to Section 3.2 discussing this: "Given the variability in the sub-grid-processed size distributions in Fig. 1, assuming a single emissions size distribution for biomass burning in coarse grid models may underestimate the variability in biomass burning size distributions between regions."*

**CHANGES MADE**

**Abstract**

Page 11, line 23: clarified external mixing is of black carbon

Page 11, line 24: same with internal mixing of black carbon

**1 Introduction**

Page 12, line 21: added Matsui, 2016 citation

**2.1 Model Overview**

10 Page 14, lines 7-11: added information about secondary organic aerosol and nucleation

Page 14 line 32 - page 15 line 1: added information about biomass burning emissions

**2.2 Biomass-burning emissions size and sub-grid coagulation in GEOS-Chem-TOMAS**

Equation 1: changed 84.56 to 84.576 as this was a typo

15 Page 17, line 31 – page 18, line 3: added information about uncertainties in the parameterization in Sakamoto et al., (2016).

Page 18, lines 6-10: introduced Figure 1, previously Figure 3, as it is applicable here.

Figure 3 was moved up to page 18 and becomes Figure 1

Page 19, lines 7-10: clarified that we are extrapolating the Sakamoto et al. (2016) fits.

Page 19, line 20: updated figure number

20 Page 19, lines 23-24: further justifying methods of using per-fire parameterization in the gridded model.

**2.3 Modeling radiative impacts of changes made to biomass burning emissions**

Page 20, line 4: clarifying that the aerosol number concentration and overall size distribution remains the same between both

DRE mixing states.

**3.1 Impact of biomass burning on aerosol mass**

Page 20, lines 14, 19, 20: updated figure number

Because Figure 3 became Figure 1, the old Figure 1 is now Figure 2.

**3.2 Ambient size distribution sensitivity to biomass burning**

Page 20, lines 14, 20: updated figure number

Page 21, line 7: updated figure number

Page 22, lines 2, 13: updated figure number

Because Figure 3 became Figure 1, the old Figure 2 is now the new Figure 3.

Page 22, lines 20-21: introduced new supplemental figure which shows the percent change in N10 and N80 due to subgrid

coagulation of biomass burning aerosol.

Page 23, line 1, 9: updated figure number

Page 23, lines 3-4: discussed new figure

Table 2: changed -214 to -124 because it was a typo

Page 24, lines 3-5: discussed variability in biomass burning size distribution beyond what we are capturing.

Removed old Figure 3 as it was moved up to be Figure 1

Page 25, lines 4-5: introduced new supplemental figure which shows the percent change in N80 for all sensitivity cases due

to subgrid coagulation of biomass burning aerosol.

Page 25, line 5: updated figure number

Page 26, lines 6, 18: updated figure number

Page 27, line 16: updated figure number

**3.3 Sensitivity of radiative effects**

Page 28, lines 5-6: clarified that figure 6 will be discussed further later in the section.

Page 28, lines 6, 7, 8: updated figure numbers and introduced new supplemental figures which show percent change in DRE

due to subgrid coagulation of biomass burning particles.

Page 30, lines 18, 19: updated figure number

Page 31, line 2: updated figure numbers

Page 31, lines 3-4: discussed new supplemental figures introduced earlier in section.

Page 31, line 14: updated figure numbers and introduced new supplemental figures which show percent change in AIE due to subgrid coagulation of biomass burning particles.

Page 32, lines 2, 20: updated figure numbers

Page 32, lines 2-6, 9-13, 15-18: replaced some statistics and their explanation with statistics that are easier to understand.

Page 32, lines 24-25: discussed new supplemental figures introduced earlier in section.

**3.4 Limitations of this study**

Page 34, line 11: added another reason that overlapping smoke plumes would lead to more sub-grid coagulation

Page 34, lines 13, 30: updated figure number

Page 34, lines 13-19: made discussion of overlapping smoke plumes more quantitative.

Page 35, lines 4, 7, 8: updated figure number

Page 35, lines 20-25: added two more limitations of the study

**4 Conclusions**

Page 36, lines 1-2: clarified which case we are discussing

Page 36, lines 3-5: Made discussion more quantitative.

Page 36, line 29: clarified that increasing the magnitude of the estimated DRE increases the cooling.

Page 37, line 5: fixed typo

**Supplement**

Added new figure S2 showing temporal variability of modeled size distribution

5   Changed old figure S2 to new figure S3

Added new figures S4 and S5 showing more information about the effect of how we adapt the per-fire parameterization to
    work for a gridded model.

Changed old figure S3 to new figure S6

Changed old figure S4 to new figure S7

10  Added new figure S8 showing more information about how sub-grid coagulation of biomass burning aerosol effects N10 and
    N80

Changed old figure S5 to new figure S9

Added new figure S10 showing more information about how sub-grid coagulation of biomass burning aerosol effects N80
    for all simulations

15  Changed old figure S6 to new figure S11

Changed old figure S7 to new figure S12

Added new figure S13 showing more information about how sub-grid coagulation of biomass burning aerosol effects DRE
    with the external mixing assumption

Changed old figure S8 to new figure S14

20  Added figure S15 showing more information about how sub-grid coagulation of biomass burning aerosol effects DRE with
    the core-shell mixing assumption

Changed old figure S9 to new figure S16

Added figure S17 showing more information about how sub-grid coagulation of biomass burning aerosol effects AIE

Changed old figure S10 to new figure S18, S11 to S19, S12 to S20, S13 to S21, and S14 to S22

[revised manuscript text omitted]

**Figure S2: Daily temporal evolution of the predicted grid-resolved median diameter ($D_{pm}$; left) and modal width (σ; right) in a box that spans 8° latitude and 10° for biomass burning emissions predicted for 2010 using FINNv1.5 fire emissions and the Sakamoto et al. (2016) parameterization after 24 hours of sub-grid coagulation with an emitted initial median diameter of 100 nm and an emitted initial modal width of 2. The top plots are centered over Alaska at 62° N, 135° W. Times that are not shown have no fire data in this gridbox. The bottom plots are centered over the Amazon at 6° S, 60° E.**

[Figure]

**Figure S2S3:** Annual-mean median diameter (a and, c) and modal width (b and, d) for biomass burning emissions predicted for 2010 using the Sakamoto et al. (2016) parameterization after 24 hours of sub-grid coagulation with an emitted initial median diameter of 100 nm and an emitted initial modal width of 2. Panels (a) and (b) show the resulting $D_{pm}$ and σ when the fire (FINNv1.5) and meteorological data is averaged over a 4°x5° grid and then that gridded data is run through the Sakamoto et al. (2016) parameterization. Panels (c) and (d) show the results when the individual fires are run through the Sakamoto et al. (2016) parameterization and then the output $D_{pm}$ and σ are averaged over a 4°x5° grid. The regions with grey cross-hatching are grid-cells with no fire data.

[Figure]

**Figure S4: Annual-mean median diameter for biomass burning emissions predicted for 2010 using the Sakamoto et al. (2016) parameterization after 24 hours of sub-grid coagulation with an emitted initial median diameter of 100 nm and an emitted initial modal width of 2. The y-axis shows the resulting median modal width when the fire (FINNv1.5) and meteorological data is averaged over a 4°x5° grid and then that gridded data is run through the Sakamoto et al. (2016) parameterization (what we use in GEOS-Chem in this study). The x-axis shows the results when the individual fires are run through the Sakamoto et al. (2016) parameterization and then the output median modal width is averaged over a 4°x5° grid. Each gridbox globally is represented as a point on the plot. The 1:1 line is in grey dash.**

[Figure]

Figure S5: Annual-mean modal width for biomass burning emissions predicted for 2010 using the Sakamoto et al. (2016) parameterization after 24 hours of sub-grid coagulation with an emitted initial median diameter of 100 nm and an emitted initial modal width of 2. The y-axis shows the resulting lognormal modal width when the fire (FINNv1.5) and meteorological data is averaged over a 4°x5° grid and then that gridded data is run through the Sakamoto et al. (2016) parameterization (what we use in GEOS-Chem in this study). The x-axis shows the results when the individual fires are run through the Sakamoto et al. (2016) parameterization and then the output lognormal modal width is averaged over a 4°x5° grid. Each gridbox globally is represented as a point on the plot. The 1:1 line is in grey dash.

[Figure]

**Figure S3S6: Effect of biomass burning on annually averaged total column OA and BC mass concentrations. The left side shows the total column mass concentration of (a) OA and (c) BC in the simulations with FINNv1 biomass burning emissions (i.e., *noSubCoag_FINN*). The right side shows the percent of the mass in the column that is due to biomass burning emissions for (b) OA and (d) BC by taking the difference between the *noSubCoag_FINN* and *noBB* simulations.**

[Figure]

**Figure S4S7: Effect of biomass burning on surface-level number concentration of particles above 10 nm (a, b, c) and 80 nm (d, e, f). Panels (a) and (d) show the absolute number concentration for the *noBB* simulation. Panels (b) and (e) show the percent increase due to FINNv1 biomass burning emissions from the *noBB* simulation to the *noSubCoag_FINN*simulation. Panels (c) and (f) show the percent increase due to FINNv1 biomass burning emissions from the *noBB* simulation to the *SubCoag_FINN* simulation. The number in the bottom right of each panel is the global mean percent increase due to biomass burning.**

[Figure]

**Figure S8: Percent change in the relative contribution of biomass burning to particles larger than 10 nm (N10) and particles larger than 80 nm (N80) due to sub-grid coagulation when using GFED emissions (left) or FINNv1 emissions (right), and an initial median diameter of 100 nm, initial modal width of 2, and coagulation time of 24 hours. Negative values correspond to a reduced impact of biomass burning when sub-grid coagulation is added. Regions where the percent increase of N10 or N80 due to biomass burning is less than 1% (with or without sub-grid coagulation, see Figs. 3 and S7) are shaded in grey.**

[Figure]

**Figure S9**: Annual-average percent changes in N80 at the surface level due to the inclusion of FINNv1 biomass burning emissions relative to the simulation without biomass burning (*noBB*). Panels (a), (c), and (e) have no sub-grid coagulation (*D150_noSubCoag_FINN*, *noSubCoag_FINN*, and *s1.6_noSubCoag_FINN*, respectively). Panels (b), (d), and (f) have sub-grid coagulation (*D150_SubCoag_FINN*, *SubCoag_FINN*, and *s1.6_SubCoag_FINN*, respectively). Panels (a) and (b) have an emitted median diameter of 150 nm and an emitted modal width of 2. Panels (c) and (d) have an emitted median diameter of 100 nm and an emitted modal width of 2. Panels (e) and (f) have an emitted median diameter of 100 nm and an emitted modal width of 1.6. The number in the bottom right of each panel is the global mean percent increase in N80 due to biomass burning.

[Figure]

**Figure S10: Percent change in the relative contribution of biomass burning to particles larger than 80 nm due to sub-grid coagulation when using GFED emissions (left) or FINNv1 emissions (right) and a coagulation time of 24 hours. Panels in the top row have an emitted median diameter of 150 nm and an emitted modal width of 2. Panels in the middle row have an emitted median diameter of 100 nm and an emitted modal width of 2. Panels in the bottom row have an emitted median diameter of 100 nm and an emitted modal width of 1.6. Negative values correspond to a reduced impact of biomass burning when sub-grid coagulation is added. Regions where the percent increase of N80 due to biomass burning is less than 1% (with or without sub-grid coagulation, see Figs. 4 and S9) are shaded in grey.**

[Figure]

**Figure S6S11: Predicted grid-resolved aerosol size distributions with FINNv1 biomass burning emissions over Alaska at 62° N, 140° W, averaged over the June, July, and August fire season (a and c) and the Amazon at 6° S, 60° E, averaged over the August, September, and October fire season (b and d). All panels show the size distributions for the noBB, noSubCoag_FINN, and SubCoag_FINN simulations in the dashed green, solid blue, and solid pink lines, respectively. The top panels (a and b) show the sensitivity to the emitted median diameter, and the bottom panels (c and d) show the sensitivity to the emitted modal width. Note the different y-axis scales.**

[Figure]

**Figure S7S12: All-sky direct radiative effect (DRE) due to biomass burning aerosols using FINNv1 emissions and using the external-mixing assumption. Panels (a), (c), and (e) are without sub-grid coagulation (*D150_noSubCoag_FINN*, *noSubCoag_FINN*, and *s1.6_noSubCoag_FINN*, respectively). Panels (b), (d), and (f) are with sub-grid coagulation(*D150_SubCoag_FINN*, *SubCoag_FINN*, and *s1.6_SubCoag_FINN*, respectively). Panels (a) and (b) have an emitted median diameter of 150 nm and an emitted modal width of 2. Panels (c) and (d) have an emitted median diameter of 100 nm and an emitted modal width of 2. Panels (e) and (f) have an emitted median diameter of 100 nm and an emitted modal width of 1.6. The number in the bottom right of each panel is the global mean DRE value [mW m$^{-2}$].**

[Figure]

**Figure S13: Percent change in all-sky direct radiative effect (DRE) of biomass burning aerosol due to sub-grid coagulation when using the external mixing assumption and GFED emissions (left) or FINNv1 emissions (right) and an assumed in-plume coagulation time of 24 hours. Panels in the top row have an emitted median diameter of 150 nm and an emitted modal width of 2. Panels in the middle row have an emitted median diameter of 100 nm and an emitted modal width of 2. Panels in the bottom row have an emitted median diameter of 100 nm and an emitted modal width of 1.6. Positive (red) values correspond to an increased cooling tendency of biomass burning DRE due to sub-grid coagulation being added. Regions in grey indicate that the DRE due to biomass burning is a more positive value than -100 mW m$^{-2}$ (i.e. there is less than 100 mW m$^{-2}$ of cooling) with or without sub-grid coagulation (see Figs. 7 and S12).**

[Figure]

**Figure S8S14: All-sky direct radiative effect (DRE) due to biomass burning aerosols using FINNv1 emissions and using the core-shell mixing assumption. Panels (a), (c), and (e) are without sub-grid coagulation (*D150_noSubCoag_FINN*, *noSubCoag_FINN*, and *s1.6_noSubCoag_FINN*, respectively). Panels (b), (d), and (f) are with sub-grid coagulation (*D150_SubCoag_FINN*, *SubCoag_FINN*, and *s1.6_SubCoag_FINN*, respectively). Panels (a) and (b) have an emitted median diameter of 150 nm and an emitted modal width of 2. Panels (c) and (d) have an emitted median diameter of 100 nm and an emitted modal width of 2. Panels (e) and (f) have an emitted median diameter of 100 nm and an emitted modal width of 1.6. The number in the bottom right of each panel is the global mean DRE value [mW m$^{-2}$].**

[Figure]

**Figure S15: Percent change in all-sky direct radiative effect (DRE) of biomass burning aerosol due to sub-grid coagulation when using the core-shell mixing and GFED emissions (left) or FINNv1 emissions (right) and an assumed in-plume coagulation time of 24 hours. Panels in the top row have an emitted median diameter of 150 nm and an emitted modal width of 2. Panels in the middle row have an emitted median diameter of 100 nm and an emitted modal width of 2. Panels in the bottom row have an emitted median diameter of 100 nm and an emitted modal width of 1.6. Positive (red) values correspond to an increased cooling tendency of biomass burning DRE due to sub-grid coagulation being added. Regions in grey indicate that the DRE due to biomass burning is a more positive value than -100 mW m$^{-2}$ (i.e. there is less than 100 mW m$^{-2}$ of cooling) with or without sub-grid coagulation (see Figs. 8 and S14).**

[Figure]

**Figure S9S16: Cloud-albedo aerosol indirect effect (AIE) due to biomass burning aerosols using FINNv1 emissions. Panels (a), (c), and (e) are without sub-grid coagulation (*D150_noSubCoag_FINN*, *noSubCoag_FINN*, and *s1.6_noSubCoag_FINN*, respectively). Panels (b), (d), and (f) are with sub-grid coagulation(*D150_SubCoag_FINN*, *SubCoag_FINN*, and *s1.6_SubCoag_FINN*, respectively). Panels (a) and (b) have an emitted median diameter of 150 nm and an emitted modal width of 2. Panels (c) and (d) have an emitted median diameter of 100 nm and an emitted modal width of 2. Panels (e) and (f) have an emitted median diameter of 100 nm and an emitted modal width of 1.6. The number in the bottom right of each panel is the global mean AIE value [mW m$^{-2}$].**

[Figure]

**Figure S17: Percent change in cloud albedo aerosol indirect effect (AIE) of biomass burning aerosol due to sub-grid coagulation and GFED emissions (left) or FINNv1 emissions (right) and an assumed in-plume coagulation time of 24 hours. Panels in the top row have an emitted median diameter of 150 nm and an emitted modal width of 2. Panels in the middle row have an emitted median diameter of 100 nm and an emitted modal width of 2. Panels in the bottom row have an emitted median diameter of 100 nm and an emitted modal width of 1.6. Positive (red) values correspond to an increased cooling tendency of biomass burning AIE due to sub-grid coagulation being added. Regions in grey indicate that the AIE due to biomass burning is a more positive value than -100 mW m$^{-2}$ (i.e. there is less than 100 mW m$^{-2}$ of cooling) with or without sub-grid coagulation (Figs. 9 and S16).**

[Figure]

**Figure S10S18: Effect of biomass burning on surface-level N10 (a-c) and N80 (d-f) under three sub-grid coagulation conditions. Panels (a) and (d) show the *noSubCoag_FINN* case (no sub-grid coagulation). Panels (b) and (e) show the *SubCoag_FINN* case (with sub-grid coagulation as in the rest of the paper, where the smoke plumes are treated as without overlap). Panels (c) and (f) show a new case where all smoke plumes in the gridbox completely overlap and form a single "superplume" upon emission into the sub-grid coagulation parameterization. All panels show the percent increase due to FINNv1 biomass burning emissions relative to the *noBB* simulation. The number in the bottom right of each panel is the global mean percent increase due to biomass burning.**

[Figure]

**Figure S11S19**: Annual-average percent changes in N80 at the surface level due to the inclusion of GFED biomass burning emissions relative to the simulation without biomass burning (*noBB*). On the left, there is no sub-grid coagulation (*noSubCoag*). In the middle, the sub-grid coagulation time is 12 hours (*SubCoag_12h*). On the right, the sub-grid coagulation time is 24 hours (*SubCoag*). The number in the bottom right corner of each panel is the global mean percent increase in N80 due to biomass burning.

[Figure]

**Figure S12S20: All-sky direct radiative effect due to biomass burning aerosols using GFED emissions and the external-mixing assumption. On the left, there is no sub-grid coagulation (*noSubCoag*). In the middle, the sub-grid coagulation time is 12 hours (*SubCoag_12h*). On the right, the sub-grid coagulation time is 24 hours (*SubCoag*). The number in the bottom right corner of each panel is the global mean value [mW m$^{-2}$].**

[Figure]

**Figure S21: All-sky direct radiative effect due to biomass burning aerosols using GFED emissions and the internal, core-shell mixing assumption. On the left, there is no sub-grid coagulation (*noSubCoag*). In the middle, the sub-grid coagulation time is 12 hours (*SubCoag_12h*). On the right, the sub-grid coagulation time is 24 hours (*SubCoag*). The number in the bottom right corner of each panel is the global mean value [mW m$^{-2}$].**

[Figure]

**Figure S14S22: Cloud-albedo aerosol indirect effect due to biomass burning aerosols using GFED emissions. On the left, there is no sub-grid coagulation (*noSubCoag*). In the middle, the sub-grid coagulation time is 12 hours (*SubCoag_12h*). On the right, the sub-grid coagulation time is 24 hours (*SubCoag*). The number in the bottom right corner of each panel is the global mean value [mW m$^{-2}$].**